# TurboReBeL: 250× Accelerated Belief Learning for large Imperfect-Information Extensive-Form Games

## Abstract

Recursive Belief-based Learning (ReBeL) provides a general framework for large-scale Imperfect-Information Extensive-Form Games (IIEFGs) by integrating self-play reinforcement learning with search. However, ReBeL suffers from prohibitive computational costs during training: each Public Belief State (PBS) sample requires $T$ iterations of Counterfactual Regret Minimization (CFR), and the PBS state space necessitates billions of samples for convergence. For example, training ReBeL on games such as Heads-Up No-Limit Texas Hold'em (HUNL) from scratch demands $4.5$ billion samples and $2$ million GPU hours. To address this, we propose *TurboReBeL*, which achieves a $\sim 250\times$ acceleration in training through two key innovations: (i) Single-Sample Multi-Iteration Generation: This core innovation fixes subgame strategies to CFR-averaged policies, generating data for all $T$ iterations in one sampling pass and yielding a theoretical $O(T)$ speedup. (ii) Isomorphic Data Augmentation: This technique enhances sample diversity through game-theoretic invariants (suit and chip isomorphism) with minimal overhead and no performance loss. Evaluations show that TurboReBeL matches ReBeL's exploitability in Turn Endgame Hold'em using approximately $0.4\%$ of the training cost, and achieves comparable performance on HUNL with $450\times$ fewer samples. TurboReBeL is the first depth-limited solving framework that combines ultra-fast training, strong scalability, low exploitability, theoretical convergence guarantees, human-data-free training, and fast real-time decision-making, representing a fundamental breakthrough in solving IIEFGs.

## 1 Introduction

Imperfect-Information Extensive-Form Games (IIEFGs) model sequential multi-agent decision-making under partial observability, providing a fundamental framework for games such as poker (Bowling et al., 2017), dark chess (Zhang & Sandholm, 2021), and Mahjong (Liu et al., 2023). Counterfactual Regret Minimization (CFR) (Zinkevich et al., 2007) and its variants (Lanctot et al., 2009; Brown & Sandholm, 2019b; Farina et al., 2021; Xu et al., 2024) have emerged as dominant solutions due to their empirical efficacy in solving IIEFGs via recursive game tree traversal. However, the computational cost of CFR scales exponentially with game depth, rendering it intractable to solve for large IIEFGs such as Heads-Up No-Limit Texas Hold'em (HUNL), a game with $\sim 10^{165}$ states (Johanson, 2013). To address this, depth-limited solving (Kroer & Sandholm, 2015) combines limited look-ahead search with neural network value approximation (Moravčík et al., 2017), enabling strong strategies with low-exploitability within seconds (Li & Huang, 2025).

Recursive Belief-based Learning (ReBeL) (Brown et al., 2020) represents a groundbreaking paradigm in this domain. In contrast to earlier approaches such as blueprint-based methods, which depend on high-quality blueprint strategies and require extensive real-time computation during game-play (Brown et al., 2018), or data-driven methods that rely on fixed, carefully curated human demonstration data with limited potential for iterative improvement (Moravčík et al., 2017), ReBeL introduces a general-purpose self-play reinforcement learning framework that learns directly from scratch. This approach achieves strong scalability, low exploitability, strong theoretical convergence guarantees, and fast real-time decision-making capabilities. Consequently, ReBeL or its underlying

concept has been widely applied in the intelligent agents of complex games, such as no-press Diplomacy (Bakhtin et al., 2021) and no-limit poker (Li et al., 2024).

However, the training cost of ReBeL is prohibitively large due to three fundamental bottlenecks: (i) *High per-sample cost*: each subgame requires $T$ full CFR iterations (typically $T = 250$) to reach strategy convergence; (ii) *Low sampling efficiency*: only one Public Belief State (PBS) data point is generated per $T$ iterations; (iii) *Massive data requirement*: the PBS value network needs to generalize across all PBSs and iterations, necessitating billions of self-play samples. These bottlenecks make ReBeL exceptionally costly to train on large-scale games such as HUNL,[1] highlighting the critical need for improved sampling efficiency.

To overcome these limitations, we introduce *TurboReBeL*. Our objective is not merely to accelerate ReBeL, but also to preserve its myriad benefits while breaking its fundamental efficiency barrier. We achieve this through a novel framework that integrates two core innovations: (i) Single-Sample Multi-Iteration Generation; (ii) Isomorphic Data Augmentation. Our motivation is to decouple the process of strategy evolution from value estimation, enabling the generation of training data for multiple CFR iterations within a single strategically consistent depth-limited subgame solving.

*Single-Sample Multi-Iteration Generation* simultaneously processes PBSs corresponding to $T$ CFR iterations after leaf node sampling. By establishing a depth-limited subgame rooted at the sampled leaf and fixing strategies to the CFR-averaged policy, we compute expected values for all $T$ PBSs which are from previous depth-limited solving in a single traversal. This innovation achieves up to $T\times$ sampling acceleration, directly addressing the primary sampling bottleneck of ReBeL. To our knowledge, this is the first method that specifically accelerates value network training in depth-limited solving, markedly enhancing the efficiency of iterative self-play frameworks. We also provide theoretical proofs for the value estimation error during depth-limited solving and convergence guarantee with the same order as in ReBeL.

*Isomorphic Data Augmentation* leverages game-theoretic symmetries (Tewolde et al., 2025) to diversify PBS samples with isomorphic public states. For poker, selected for its inherent structural symmetries, strategic complexity and historical significance (Brown & Sandholm, 2018; 2019a), we implement: (i) *Suit Isomorphism*: remapping card suits via $24$ distinct transformations (Gilpin & Sandholm, 2006); and (ii) *Chip Isomorphism*: scaling chip counts by multiplicative factors. Critically, this augmentation operates directly on collected PBS data without re-running costly CFR computations, introducing no estimation error. To our knowledge, this is the first method to incorporate data augmentation into depth-limited solving frameworks such as ReBeL.

To illustrate the acceleration achieved by TurboReBeL, we conduct evaluations on two large IIEFGs: Turn Endgame Hold'em (TEH) and Heads-up No-Limit Texas Hold'em (HUNL). On TEH, TurboReBeL achieves an exploitability comparable to ReBeL with approximately $0.4\%$ of the training cost. On HUNL, despite using $450\times$ fewer samples than the $4.5$ billion needed for ReBeL, TurboReBeL achieves a competitive win-rate of $42 \pm 12$ mbb/hand against Slumbot (Jackson, 2013), comparable to the $45 \pm 5$ mbb/hand achieved by ReBeL (Brown et al., 2020). These results collectively demonstrate a $\sim 250\times$ training acceleration without performance loss.

TurboReBeL establishes a new general depth-limited solving framework characterized by its breakthrough training efficiency. It fully maintains the key strengths of prior work: low exploitability, theoretical convergence guarantees, human-data-free training, and real-time solving speed. Our work makes three fundamental contributions:

- *Sampling Acceleration Framework*: We introduce a novel single-sample multi-iteration generation approach that fixes subgame strategies to CFR-averaged policies, enabling the generation of PBS data for all $T$ iterations in one sampling. This yields an $O(T)$ speedup and directly alleviates the core sampling bottleneck of ReBeL.

- *Belief-Invariant Augmentation*: We develop novel isomorphic transformations, including suit isomorphism and chip isomorphism, for the depth-limited solving framework. These game-theoretic invariants increase sample diversity while preserving belief state consistency, without requiring additional CFR computations.

---

[1]Training ReBeL on HUNL from scratch requires over 4.5 billion samples and 2 million GPU hours (Brown et al., 2020).

- *Training Efficiency Improvement*: We demonstrate a practical framework, *TurboReBeL*, which reduces ReBeL's training cost by $\sim 250\times$ while maintaining strong performance. This enables large-scale IIEFG training with dramatically lower resources, opening new possibilities for research in efficient depth-limited solving.

## 2 RELATED WORK

**Depth-limited Solving in IIEFGs.** A predominant line of work in depth-limited solving estimates dynamic values during CFR iterations. DeepStack (Moravčík et al., 2017) established this direction, attaining superhuman HUNL performance via real-time belief estimation and heuristic sampling, albeit while being fundamentally constrained by its use of static data, which impedes continuous learning and caps ultimate performance. Recursive Belief-based Learning (ReBeL) (Brown et al., 2020) constituted a significant advance by formalizing a self-play reinforcement learnining procedure that learns value networks from scratch and iteratively improves its training data. The more general Student of Games (SoG) (Schmid et al., 2023) unified search and learning but sacrifices specialization, resulting in lower performance than ReBeL in large games such as HUNL.

An alternative methodology pre-estimates value functions from a set of blueprint strategies before running CFR. For instance, Modicum (Brown et al., 2018) relies on expensive pre-trained blueprint strategies, whereas Pluribus (Brown & Sandholm, 2019a) employed self-play-based blueprint strategies for faster execution. SePoT (Kubícek et al., 2024) further abstracted blueprint strategies into model representations, scaling to large public state spaces. Although blueprint-based methods reduce auxiliary training overhead, they require substantial real-time computation to evaluate multiple strategies and, more critically, depend heavily on the quality of blueprint strategies (Kovarík et al., 2023), which are difficult to obtain effectively under limited training time.

TurboReBeL distinguishes itself by drastically reducing the prohibitive training cost of ReBeL, while avoiding the extensive real-time computation and blueprint dependency of alternative approaches. It retains the self-play, from-scratch learning paradigm of ReBeL but introduces fundamental innovations in the data generation process to achieve dramatically greater efficiency.

Recent refinements in depth-limited solving continue to enhance real-time performance. RL-CFR (Li et al., 2024) employed reinforcement learning for adaptive action abstraction, and EVPA (Li & Huang, 2025) accelerated real-time solving through online pruning and abstraction techniques. However, these methods primarily optimize decision-time performance and still rely on foundation models requiring extensive offline training. TurboReBeL directly addresses this bottleneck by providing a highly sample-efficient foundation model that drastically reduces pre-training overhead. As such, it can serve as a scalable base for these refinements, enabling them to achieve high real-time performance with significantly lower training cost.

**Sample Efficiency in Games.** Improving sample efficiency remains a central challenge in reinforcement learning. In perfect-information settings, EfficientZero (Ye et al., 2021) achieved superhuman performance on Atari with a fraction of the typical data requirement by combining MCTS-based reinforcement learning with efficient search. Similarly, in complex imperfect-information domains such as Mahjong, techniques such as variance reduction have been employed to accelerate training (Li et al., 2022). The IRIS agent (Micheli et al., 2023) leveraged a transformer-based world model to improve sample efficiency in long-horizon tasks. In multi-agent settings, curriculum learning has been shown to expedite convergence in zero-sum games (Chen et al., 2024).

Within the specific context of CFR, most prior work on sample efficiency has focused on extensive-form sampling for full-game, rather than depth-limited solving. Deep CFR (Brown et al., 2019) utilized neural function approximation to reduce the number of traversals needed, DREAM (Steinberger et al., 2020) employed importance sampling to reduce variance, and ESCHER (McAleer et al., 2023) forewent importance sampling in favor of a learned value function for regret estimation. However, these methods do not support iterative strategy refinement within subgames. Consequently, they yield coarser policies with higher exploitability compared to subgame solving methods. TurboReBeL represents the first method to specifically address sampling efficiency within the depth-limited solving paradigm. It preserves ReBeL's real-time subgame solving capability while maintaining the ability to compute highly refined, low-exploitability strategies, while achieving orders-of-magnitude improvement in data efficiency.

## 3 BACKGROUND AND NOTATION

An Imperfect-Information Extensive-Form Game (IIEFG) is formally defined by the tuple $G = (\mathcal{N}, \mathcal{H}, \mathcal{A}, \mathcal{Z}, \mathcal{P}, u, \sigma_c, \mathcal{I})$. Here, $\mathcal{N} = \{1, \ldots, N\}$ denotes the set of players. The set $\mathcal{H}$ contains all possible histories (states), each expressed as a sequence of actions starting from the initial history $\emptyset$. For any history $h \in \mathcal{H}$, the set of available actions is denoted by $\mathcal{A}(h)$. Executing an action $a \in \mathcal{A}(h)$ transitions to a new history $ha$. A history $h$ is an ancestor of $h'$ (denoted $h \sqsubseteq h'$) if $h'$ is reachable from $h$. The set $\mathcal{Z} \subseteq \mathcal{H}$ denotes terminal histories. For any terminal history $z \in \mathcal{Z}$, $u_p(z)$ represents the payoff for player $p$ at $z$. The player function $\mathcal{P} : \mathcal{H} \setminus \mathcal{Z} \to \mathcal{N} \cup \{c\}$ specifies who acts at a non-terminal history $h$, where $c$ denotes the chance player that follows a fixed strategy $\sigma_c(h, a)$. Imperfect information is represented by information sets: for each player $p$, $\mathcal{I}_p \in \mathcal{I}$ partitions the histories indistinguishable to $p$. For any $I_p \in \mathcal{I}_p$ and $h, h' \in I_p$, player $p$ cannot distinguish between $h$ and $h'$. When $p$ is the acting player at $I_p$, the information set could be denoted simply as $I$.

The behavioral strategy for player $p$ is denoted $\sigma_p$, which assigns to each information set $I \in \mathcal{I}_p$ a probability distribution over actions in $\mathcal{A}(I)$. The probability of choosing action $a$ at $I$ is denoted $\sigma(I, a)$. A strategy profile $\sigma = (\sigma_1, \ldots, \sigma_N)$ consists of strategies for all players. The probability of reaching history $h$ under $\sigma$ is $\pi^\sigma(h)$, which can be factored into the contribution of player $p$, denoted $\pi_p^\sigma(h)$, and the contribution of all other players and chance, denoted $\pi_{-p}^\sigma(h)$. Under a strategy profile $\sigma$, the expected utility for player $p$ at a given history $h$ is expressed as $u_p^\sigma(h)$, and the overall expected utility in the full game is denoted $u_p(\sigma)$. The best response of player $p$ to $\sigma_{-p}$ is $BR(\sigma_{-p}) = \arg\max_{\sigma_p} u_p(\sigma_p, \sigma_{-p})$. A Nash equilibrium $\sigma^*$ satisfies $\forall p, u_p(\sigma^*) = \max_{\sigma_p} u_p(\sigma_p, \sigma_{-p}^*)$. The exploitability (Davis et al., 2014) of a strategy profile $\sigma$ is given by $\text{Exploitability}(\sigma) = \left(\sum_{p \in \mathcal{N}} u_p(BR(\sigma_{-p}), \sigma_{-p})\right)/|\mathcal{N}|$.

The Counterfactual Value (CFV) for player $p$ at information set $I_p$ under strategy profile $\sigma$ is calculated as $v_p^\sigma(I_p) = \sum_{h \in I_p} \pi_{-p}^\sigma(h) \sum_{z \in \mathcal{Z}, h \sqsubseteq z} \pi^\sigma(z \mid h) u_p(z)$. Counterfactual Regret Minimization (CFR) (Zinkevich et al., 2007) is an iterative algorithm for approximating Nash equilibrium. Let $\sigma^t$ be the strategy profile at iteration $t$. The instantaneous regret for action $a$ at $I$ is $r^t(I, a) = v_p^{(\sigma^t(I \to a), \sigma_{-p}^t)}(I) - v_p^{\sigma^t}(I)$, where $\sigma^t(I \to a)$ denotes a strategy profile identical to $\sigma^t$ except that action $a$ is always taken at $I$. The cumulative regret is $R^t(I, a) = \sum_{t'=1}^t r^{t'}(I, a)$. The strategy is updated via regret matching (Hart & Mas-Colell, 2000): $\sigma^{t+1}(I, a) = R_+^t(I, a)/(\sum_{a' \in \mathcal{A}(I)} R_+^t(I, a'))$, where $R_+^t(I, a) = \max(0, R^t(I, a))$ and the strategy is arbitrary if the denominator is zero. The average strategy $\overline{\sigma}^T(I, a)$ is computed as $\overline{\sigma}^T(I, a) = (\sum_{t=1}^T (\pi_p^{\sigma^t}(I) \cdot \sigma^t(I, a)))/(\sum_{t=1}^T (\pi_p^{\sigma^t}(I)))$.

A public state (node) $s$ encapsulates all publicly observable information. For each player $p$, $\mathcal{I}_p(s)$ denotes the set of their information sets consistent with $s$. A subgame $S$ is a connected subtree of the full game tree. A node $s \in S$ is a leaf (respectively, root) of $S$ if it has no descendants (ancestors) within $S$. A Public Belief State (PBS) $\beta_s$ at public state $s$ is a profile of belief distribution over the possible information sets of all players that are consistent with the public state $s$ (Nayyar et al., 2013). Formally, $\beta_s = (b_1, \ldots, b_N)$, where each $b_p \in \Delta(\mathcal{I}_p(s))$ is a distribution over $\mathcal{I}_p(s)$. The CFV for player $p$ at information set $I_p \in \mathcal{I}_p(s)$ under strategy profile $\sigma$, given the PBS $\beta_s$, is $v_p^\sigma(I_p \mid \beta_s) = \sum_{h \in I_p} \pi^\sigma(h \mid I_p, \beta_s) \cdot v_p^\sigma(h)$, where $\pi^\sigma(h \mid I_p, \beta_s)$ is the probability of reaching history $h$ given that player $p$'s information set is $I_p$ and the beliefs over other players' information sets are specified by $\beta_s$. The PBS value vector under $\sigma$ is denoted $\mathbf{v}^\sigma(\beta_s) = \left(v_p^\sigma(I_p \mid \beta_s)\right)_{p \in \mathcal{N}, I_p \in \mathcal{I}_p(s)}$.

## 4 ReBeL FRAMEWORK

This section outlines the Recursive Belief-based Learning (ReBeL) framework (Brown et al., 2020), a general self-play reinforcement learning method for solving large IIEFGs. ReBeL operates through two primary phases: training and sampling. During training, a value network parameterized by $\theta$ is learned to approximate the CFV for any given PBS. Further details regarding the network architecture and hyperparameters are provided in Appendix B.

Algorithm 1 presents the sampling process of the ReBeL algorithm. The process begins at the initial PBS $\beta_{\text{init}}$. For each sampled PBS $\beta_s$, a depth-limited subgame $S$ is constructed with $\beta_s$ as its root. Subsequently, $T$ iterations of CFR are executed within $S$. In each iteration $t$:

- Starting from $\beta_s$ and using the average strategy $\overline{\sigma}^{t-1}$, the PBS $\beta_{s_{\text{leaf}}}^{\overline{\sigma}^{t-1}}$ is computed for every leaf node $s_{\text{leaf}} \in S$;

- The PBS value network $\hat{\mathbf{v}}^\theta(\beta_{s_{\text{leaf}}}^{\overline{\sigma}^{t-1}})$ provides CFV estimates for all information sets at each leaf node $s_{\text{leaf}} \in S$;

- These CFVs are then backed up from leaves to the root according to $\sigma^{t-1}$, updating the instantaneous regrets $r^t(I, a)$, cumulative regrets $R^t(I, a)$, and the current strategy $\sigma^t(I, a)$;

- The average strategy $\overline{\sigma}^t$ and PBS value vector $\mathbf{v}^{\overline{\sigma}}(\beta_s)$ are updated incrementally.

---

**Algorithm 1** ReBeL (Brown et al., 2020): Sampling process for PBS $\beta_s$

---

1: **function** SAMPLING($\beta_s, \theta$)
2:     $S \leftarrow$ BuildTree($\beta_s$)
3:     $\overline{\sigma}^0, \sigma^0 \leftarrow$ UniformPolicy($S$), $\mathbf{v}^{\overline{\sigma}}(\beta_s) \leftarrow \mathbf{0}$
4:     **for** $t = 1$ **to** $T$ **do**
5:         $S \leftarrow$ UpdateLeafCFV($S, \overline{\sigma}^{t-1}, \theta$)       ▷ Update leaf node CFVs using value network
6:         $\sigma^t \leftarrow$ CFRSolving($S, \sigma^{t-1}$)    ▷ Perform CFR iteration to update regrets and strategies
7:         $\overline{\sigma}^t \leftarrow \frac{t}{t+1}\overline{\sigma}^{t-1} + \frac{1}{t+1}\sigma^t$
8:         $\mathbf{v}^{\overline{\sigma}}(\beta_s) \leftarrow \frac{t}{t+1}\mathbf{v}^{\overline{\sigma}}(\beta_s) + \frac{1}{t+1}\mathbf{v}^{\sigma^t}(\beta_s)$
9:     **end for**
10:   Add $\{\beta_s, \mathbf{v}^{\overline{\sigma}}(\beta_s)\}$ to *Data*
11:   $t_{\text{next}} \sim$ Uniform$\{0, T\}$, $s_{\text{leaf}} \leftarrow$ SampleLeaf($S, \sigma^{t_{\text{next}}}$)
12:   **if** $s_{\text{leaf}}$ is not terminal **then**
13:     **return** Sampling($\beta_{s_{\text{leaf}}}^{\overline{\sigma}^{t_{\text{next}}}}, \theta$)    ▷ Recursively sample from non-terminal leaf node PBS
14:   **end if**
15: **end function**

---

After completing $T$ iterations, the pair $\{\beta_s, \mathbf{v}^{\overline{\sigma}}(\beta_s)\}$ is stored as training data. The algorithm then randomly selects an iteration $t_{\text{next}}$ and samples a leaf node $s_{\text{leaf}}$ from the subgame. If $s_{\text{leaf}}$ is not a terminal public state, the sampling process continues recursively from the resulting PBS $\beta_{s_{\text{leaf}}}^{\overline{\sigma}^{t_{\text{next}}}}$.

ReBeL provides strong performance with theoretical guarantees, fast real-time solving speed, high generality, and eliminating the need for human data or domain-specific knowledge in large IIEFGs. As a result, ReBeL or its idea has been applied in the intelligent agents of complex imperfect-information games, such as no-press Diplomacy (Bakhtin et al., 2021) and no-limit poker (Li et al., 2024). Despite these strengths, the practical applicability of ReBeL is severely limited by three computational bottlenecks that lead to prohibitive training costs.

First, the *high per-sample cost* arises from the CFR solving time complexity of $O(T|S||I|)$, where $|S|$ is the number of public states in the depth-limited subgame and $|I|$ is the number of information sets per public state for all players. With typical HUNL parameters ($T = 250$, $|S| \approx 200$, $|I| = 2,652$), each sample requires $\sim 10^8$ operations, amounting to roughly 2 seconds of solving per subgame (Brown et al., 2020).

Second, ReBeL exhibits inherently *low sampling efficiency*. Although each sampling procedure involves substantial computation including subgame construction, CFR iterations, and value estimations, only the root PBS and its value vector after $T$ iterations are retained as a single training example. All intermediate computational results, such as evolving strategies and belief states from earlier iterations, are discarded, leading to inefficient data utilization.

Third, the method requires *massive training data*. The value network needs to accurately approximate CFVs across diverse information sets under varying PBSs. Furthermore, since ReBeL is trained from scratch through self-play, it needs to gradually learn to model the approximately equilibrium PBS distribution throughout training. These factors collectively necessitate an enormous number of samples. For instance, the original implementation of ReBeL required 4.5 billion samples to train on HUNL from scratch (Brown et al., 2020).

These bottlenecks collectively result in extreme computational demands: training ReBeL from scratch on HUNL requires approximately 2 million GPU hours using 32GB Nvidia V100 GPUs (Brown et al., 2020). Such costs are already prohibitive and will become intractable in larger games, such as multi-player no-limit Hold'em (Brown & Sandholm, 2019a). It is important to note that these limitations are inherent to the depth-limited solving paradigm, not unique to ReBeL. Any previous depth-limited solving method will face similar scalability challenges, highlighting the critical need for fundamental improvements in sampling efficiency and underscoring the significance of our work in overcoming these barriers.

## 5 TURBOREBEL FRAMEWORK

This section introduces TurboReBeL, a framework that tackles ReBeL's fundamental efficiency bottlenecks. As illustrated in Figure 1, our key innovation is to decouple strategy evolution from value estimation, allowing a single traversal to generate training data for multiple CFR iterations. TurboReBeL integrates two core techniques during sampling: (i) Single-Sample Multi-Iteration Generation (Section 5.1), generating PBS data for all $T$ iterations in a CFR depth-limited solving; and (ii) Isomorphic Data Augmentation (Section 5.2), boosting data diversity without approximation error.

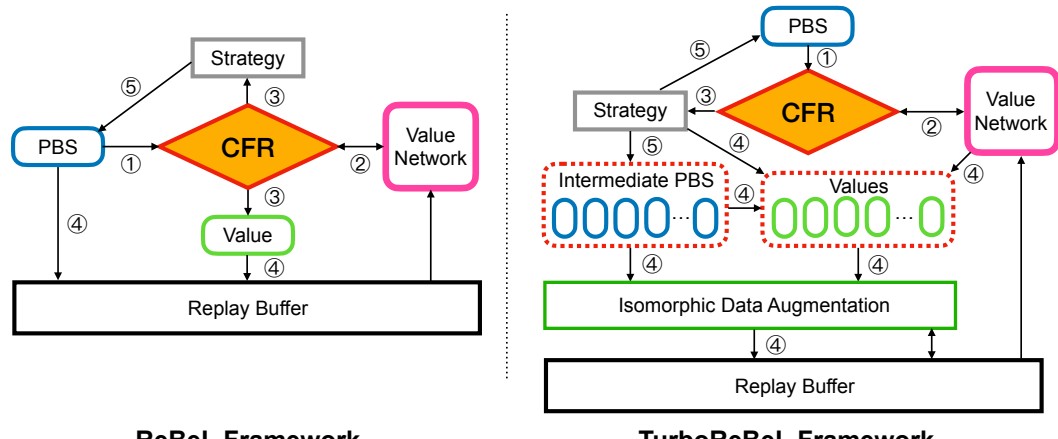

Figure 1: Schematic comparison of ReBeL (left) and TurboReBeL (right) sampling frameworks. *Steps:* ① Build a depth-limited subgame rooted at current PBS; ② Solve subgame using CFR with value network approximation; ③ Obtain strategy profile from CFR; ④ Generate PBS state-value vector data; ⑤ Sample next PBS for recursive expansion. *TurboReBeL's Key Innovation:* In Step ④, batch generation of value data for all PBSs from previous depth-limtied solving is performed under the reference strategy, decoupling value estimation from strategy evolution. Combined with isomorphic data augmentation, this yields a dramatic efficiency gain.

### 5.1 SINGLE-SAMPLE MULTI-ITERATION GENERATION

The standard ReBeL framework suffers from a critical inefficiency: it requires $T$ complete CFR iterations of a depth-limtied subgame to generate a single training example $\{\beta_s, \mathbf{v}^{\overline{\sigma}}(\beta_s)\}$. This approach is computationally prohibitive, as each iteration involves expensive subgame construction, value network queries, and regret updates.

Our key insight is that, although the sequential strategy updates in CFR are essential for regret minimization, they are largely redundant for training the value network. The value network primarily needs to learn the mapping from any PBS $\beta_s$ to its value under the full-game approximate equilibrium strategy, rather than capturing perfect PBS values in dynamically evolving subgames.

TurboReBel builds upon this insight. Specifically, during sampling, TurboReBeL recursively processes depth-limited subgames. Starting from a sampled leaf node $s$, we construct a sequence of public belief states $\beta_{s,0}, \ldots, \beta_{s,T}$ generated during the local CFR process. Each $\beta_{s,t}$ corresponds to

the game state distribution induced by the $t$-th CFR iteration within the *previous* depth-limited subgame. When TurboReBeL moves to the current depth-limited subgame rooted at $s$, we use the global averaged CFR strategy $\overline{\sigma}^T$ as the fixed reference strategy for all public belief states $\{\beta_{s,0}, \ldots, \beta_{s,T}\}$ encountered from previous depth-limited solving. In this way, all intermediate PBSs used in the current solve actually originate from the preceding subgame's CFR iterations.

Hence, for a newly encountered subgame rooted at $s$, once the reference strategy $\overline{\sigma}^T$ is computed and fixed, we can efficiently estimate the counterfactual values for all intermediate belief states $\{\beta_{s,0}, \ldots, \beta_{s,T}\}$ without CFR solving using this same fixed reference strategy. This enables the generation of $T + 1$ data points in a single depth-limited subgame solving, avoiding the need for $T + 1$ separate costly CFR solving procedures.

Algorithm 2 presents the complete TurboReBeL sampling procedure. The algorithm takes $\beta_{s,\text{avg}}$, $\{\beta_{s,0}, \ldots, \beta_{s,T}\}$ and $\theta$ as input, where $\beta_{s,\text{avg}}$ is the PBS at the root of the subgame under the reference average strategy, $\{\beta_{s,0}, \ldots, \beta_{s,T}\}$ represents the set of PBSs for each iteration $t$ that will occur during leaf node value estimation, and $\theta$ contains the parameters of the value network.

The algorithm proceeds in two distinct phases: *Strategy Evolution* and *Data Generation*.

*Phase-1: Strategy Evolution.* This phase performs standard CFR for $T$ iterations to compute the average strategy $\overline{\sigma}$ for the depth-limited subgame. This process is identical to ReBeL and ensures a high-quality reference strategy profile for subsequent value estimation. The computational cost of this phase is equivalent to one complete ReBeL sampling.

*Phase-2: Data Generation.* This phase generates training data for all $T + 1$ iterations in a single depth-limited subgame solving. For each $\beta_{s,t}$ that originated from the previous subgame's local CFR process, we construct a depth-limited subgame rooted at PBS $\beta_{s,t}$, estimate leaf values using our value network under the fixed average strategy $\overline{\sigma}$, then backpropagate these values through the subgame using $\overline{\sigma}$ to compute the root value $\mathbf{v}^{\overline{\sigma}}(\beta_{s,t})$. The pair $\{\beta_{s,t}, \mathbf{v}^{\overline{\sigma}}(\beta_{s,t})\}$ is stored as training data. This phase represents our core innovation, enabling meaningful value estimation through reference strategies whilst avoiding computationally intensive CFR solving.

---

**Algorithm 2** TurboReBeL: Single-Sample Multi-Iteration Generation

1: **function** SAMPLING($\beta_{s,\text{avg}}, \{\beta_{s,0}, \ldots, \beta_{s,T}\}, \theta$)
2:     $S \leftarrow \text{BuildTree}(\beta_{s,\text{avg}})$              ▷ Subgame rooted at reference PBS $\beta_{s,\text{avg}}$
3:     $\overline{\sigma}^0, \sigma^0 \leftarrow \text{UniformPolicy}(S)$
4:     **for** $t = 1$ **to** $T$ **do**              ▷ Phase 1: Standard ReBeL to compute $\overline{\sigma}$
5:         $S \leftarrow \text{UpdateLeafCFV}(S, \overline{\sigma}^{t-1}, \theta)$
6:         $\sigma^t \leftarrow \text{CFRSolving}(S, \sigma^{t-1})$
7:         $\overline{\sigma}^t \leftarrow \frac{t}{t+1}\overline{\sigma}^{t-1} + \frac{1}{t+1}\sigma^t$
8:     **end for**
9:     $\overline{\sigma} \leftarrow \overline{\sigma}^T$                                 ▷ Final reference strategy
10:     **for** $t = 0$ **to** $T$ **do**             ▷ Phase 2: Generate data for all iterations
11:         $S' \leftarrow \text{BuildTree}(\beta_{s,t})$        ▷ Subgame rooted at intermediate PBS $\beta_{s,t}$
12:         $S' \leftarrow \text{UpdateLeafCFV}(S', \overline{\sigma}, \theta)$    ▷ Estimate leaf values using reference strategy $\overline{\sigma}$
13:         $\mathbf{v}^{\overline{\sigma}}(\beta_{s,t}) \leftarrow \text{UpdateCFV}(S', \overline{\sigma})$   ▷ Backpropagate values with reference strategy $\overline{\sigma}$
14:         Add $\{\beta_{s,t}, \mathbf{v}^{\overline{\sigma}}(\beta_{s,t})\}$ to *Data*              ▷ Store training example
15:     **end for**
16:     $s_{\text{leaf}} \leftarrow \text{SampleLeaf}(S, \overline{\sigma})$
17:     **if** $s_{\text{leaf}}$ is not terminal **then**
18:         **return** Sampling($\beta_{s_{\text{leaf}}}^{\overline{\sigma}}, \{\beta_{s_{\text{leaf}}}^{\overline{\sigma}^0}, \ldots, \beta_{s_{\text{leaf}}}^{\overline{\sigma}^T}\}, \theta$)
19:     **end if**
20: **end function**

---

After generating all $T + 1$ data points, the algorithm samples a leaf node $s_{\text{leaf}}$ according to $\overline{\sigma}$. If non-terminal, it recursively proceeds with the new PBSs. This recursive sampling ensures that each new subgame starts from a PBS produced by the preceding subgame's local CFR solving, with the same global reference strategy $\overline{\sigma}$ held fixed for consistency across all depth levels.

Crucially, Phase-2 incurs significantly lower computational overhead compared to Phase-1. By reusing the precomputed average strategy $\overline{\sigma}$, Phase-2 avoids expensive regret calculations and strategy updates, reducing the computational burden to belief propagation and value backpropagation under a fixed policy. Empirically, Phase-2 requires less than half the computation time of Phase-1. This efficiency gain is fundamental to TurboReBeL's overall speedup, enabling the generation of $T + 1$ data points at marginal additional cost.

This approach achieves an $O(T)$ reduction in sampling cost, as the computationally intensive CFR process (Phase-1) is performed only once per subgame while yielding $T + 1$ training data points. The key innovation is using the same fixed average strategy $\overline{\sigma}$ to estimate values for all intermediate belief states, rather than performing separate CFR traversals for each iteration. To our knowledge, this is the first method that specifically accelerates value network training in depth-limited solving, markedly enhancing iterative self-play framework efficiency.

We provide a theoretical analysis of TurboReBeL, demonstrating that using the fixed average strategy $\overline{\sigma}$ for value estimation preserves ReBeL's convergence guarantees (see Appendix D for proofs).

**Theorem 1** (Value Estimation Consistency). *Let $T$ be the total CFR iteration number of TurboReBeL and $\overline{\sigma}^T$ be the reference strategy for the full game. During the depth-limited subgame solving period in the TurboReBeL, let $\overline{\sigma}^t$ be the average strategy in this depth-limited subgame at iteration $t$ of CFR ($t \leq T$). Let $s$ be an arbitrary leaf node in the depth-limited subgame, and let $\beta_{s,t}$ be any leaf PBS reached according to $\overline{\sigma}^t$. Let $\sigma^*$ be a Nash equilibrium strategy for full game. The value network output for any information set $I_p \in s$ is $\hat{v}_p^\theta(I_p \mid \beta_{s,t})$ and we set the maximum estimation error of the neural network as $\varepsilon_{\mathrm{approx}} := \sup_{I_p \in \beta_{s,t}} \left| \hat{v}_p^\theta(I_p \mid \beta_{s,t}) - v_p^{\overline{\sigma}^T}(I_p \mid \beta_{s,t}) \right|$. During depth-limited subgame solving at iteration $t$, the value estimation error for any information set $I_p \in s$ satisfies:*

$$\left| v_p^\theta(I_p \mid \beta_{s,t}) - v_p^{\sigma^*}(I_p) \right| \leq \varepsilon_{approx} + O\left(1/\sqrt{t}\right).$$

Theorem 1 provides crucial theoretical grounding for generating additional training data within depth-limited solving framework, aiming to achieve self-play reinforcement learning with fewer samples. It is important to note that the error value here refers to the fitting error to the Nash equilibrium during depth-limited solving, not the error of the training data itself. The training objective of TurboReBeL is the CFV vector of the average strategy $\overline{\sigma}^T$ given any PBS, which is consistent throughout the framework.

**Theorem 2** (Convergence of TurboReBeL). *Suppose for every depth-limited subgame, the PBS value network provides CFV estimates, whose error satisfies $\left| \hat{v}_p^\theta(I_p) - v_p^{\sigma^*}(I_p) \right| \leq \varepsilon_{\mathrm{approx}} + O\left(\frac{1}{\sqrt{t}}\right)$ for any information set used in iteration $t \leq T$. Then, the exploitability of the strategy that applies $T$-iteration CFR subgame solving in TurboReBeL is bounded by*

$$\mathrm{Expl}(\overline{\sigma}^T) = O\left(\varepsilon_{\mathrm{approx}} + \frac{1}{\sqrt{T}}\right).$$

The single-sample multi-iteration generation framework provides both practical efficiency and theoretical soundness for TurboReBeL. By simultaneously generating PBS data for all $T$ iterations through a fixed average strategy, our method achieves a theoretical $O(T)$ sampling acceleration while maintaining the $O\left(1/\sqrt{T} + \epsilon_{\mathrm{approx}}\right)$ convergence rate of standard ReBeL.

## 5.2 ISOMORPHIC DATA AUGMENTATION

While the Single-Sample Multi-Iteration Generation technique significantly enhances sampling efficiency by producing $T$ distinct PBSs from a single sampling operation, these PBSs share identical public state information despite varied belief distributions. To address this limitation and further increase training data diversity, we introduce *Isomorphic Data Augmentation*, a novel methodology that applies game-theoretically invariant transformations (Tewolde et al., 2025) to generate strategically equivalent yet syntactically distinct public states. This technique ensures that each batch

of $T$ generated PBSs varies not only in belief distributions but also in public state representations, substantially enriching the training dataset without compromising strategic validity.

To our knowledge, this represents the first successful incorporation of data augmentation into depth-limited solving frameworks such as ReBeL. We implement this approach in poker, well-suited due to its inherent structural symmetries, strategic complexity, and established role in IIEFG research (Brown & Sandholm, 2018; 2019a). Critically, the principle of isomorphic data augmentation is not limited to poker, it can be similarly defined for any game with known strategic symmetries such as dark chess and Majhong. Our method leverages two distinct forms of isomorphic transformations: Suit Isomorphism and Chip Isomorphism.

*Suit Isomorphism* capitalizes on the strategic equivalence of card suits in poker (Gilpin & Sandholm, 2006). We systematically apply all $4! = 24$ valid suit permutations to PBSs while maintaining identical strategic properties. For example, consider a PBS with public cards $(J\clubsuit, 9\clubsuit, 7\heartsuit, 2\diamondsuit)$ and a player belief distribution including hands such as $\{K\spadesuit Q\spadesuit : 0.3, A\heartsuit T\heartsuit : 0.2, 8\clubsuit 6\clubsuit : 0.5\}$. Applying a permutation mapping $\clubsuit \rightarrow \spadesuit, \spadesuit \rightarrow \heartsuit, \heartsuit \rightarrow \diamondsuit, \diamondsuit \rightarrow \clubsuit$ transforms the public cards to $(J\spadesuit, 9\spadesuit, 7\diamondsuit, 2\clubsuit)$ and the belief set to $\{K\heartsuit Q\heartsuit : 0.3, A\diamondsuit T\diamondsuit : 0.2, 8\spadesuit 6\spadesuit : 0.5\}$. This transformation yields a new PBS while preserving all strategic characteristics and equilibrium properties, as hand strengths and relative suit relationships remain identical.

*Chip Isomorphism* represents a novel contribution that enhances the inherent ability of ReBeL to handle variable stack sizes. We sample scaling factors $\alpha$ uniformly from $[0.75, 1.25]$ and multiply all chip values (stacks and pot), preserving critical pot-to-stack ratios.[2] This method produces training data covering a continuous range of stack depth scenarios, crucially important in no-limit games where strategic decisions heavily depend on effective stack sizes.

These transformations provide exceptional computational efficiency and exact strategic preservation. The operations require only linear time relative to the PBS data structure size and introduce no approximation error, maintaining the theoretical integrity of the learning process. Most importantly, these transformations preserve Nash equilibrium properties: any equilibrium strategy in the original game remains an equilibrium in the transformed game, and vice versa.

Integrated within the TurboReBeL, the augmentation process is applied during replay buffer storage. For each generated PBS, we apply $K$ randomly generated transformations from the suit and chip isomorphism sets, effectively multiplying distinct training examples by a factor of $K$. This elevates total data efficiency to $T \times K$ compared to standard ReBeL. Furthermore, we periodically reapply transformations to samples already in the replay buffer during idle training cycles, enabling continuous data renewal and diversity enhancement.

The combination of Single-Sample Multi-Iteration Generation and Isomorphic Data Augmentation provides TurboReBeL with unprecedented sampling efficiency. TurboReBeL achieves a $T\times$ improvement through the core innovation and an additional $K\times$ enhancement through data augmentation, resulting in an overall $O(TK)$ improvement over standard ReBeL. This dramatic acceleration enables practical training of large-scale IIEFGs that was previously computationally prohibitive.

## 6 EXPERIMENTS

We evaluate TurboReBeL on large IIEFGs including Heads-Up No-Limit Texas Hold'em (HUNL) and Turn Endgame Hold'em (TEH), a simplified variant of HUNL where both players are constrained to check/call during pre-flop and flop stages (see Appendix A for detailed rules). Our experimental setup closely follows the benchmarks established by ReBeL (Brown et al., 2020). All experiments use Discounted CFR (Brown & Sandholm, 2019b) with $T = 250$ iterations. Consistent with ReBeL, TurboReBeL handles opponent off-tree actions through dynamic subgame expansion during real-time decision-making and applies safe subgame solving (Moravcik et al., 2016; Brown & Sandholm, 2017). Experiments were run on a server equipped with 4 NVIDIA A100 GPUs. By leveraging established HUNL acceleration techniques such as vectorized computation and optimized payoff matrix operations (Johanson et al., 2011; 2012), our poker agents achieved an average sampling and real-time decision time of approximately one second per move, demonstrating its prac-

---

[2]We normalize players' chips to multiples of the big blind, and we allow non-integer bets during training, so this data augmentation does not result in performance loss.

ticality for real-time applications. We use $K = 2$ isomorphic data augmentation in TurboReBeL to achieve a trade-off between sampling efficiency and training time.

We performed a comprehensive evaluation of training efficiency on TEH, comparing TurboReBeL against ReBeL (Brown et al., 2020) and DeepStack (Moravčík et al., 2017) implementations over a training period of $1,000$ hours. We compare the exploitability of different algorithms under identical servers, configurations and training times. We incorporated isomorphic data augmentation in both ReBeL and DeepStack, improving their data efficiency by approximately $24\times$ compared to their original versions. As illustrated in Figure 2, DeepStack initially outperforms ReBeL but plateaus as training progresses. Both TurboReBeL and ReBeL exhibit sustained performance improvements with extended training. TurboReBeL reaches an exploitability below 0.003 in just 40 hours, while ReBeL even with isomorphic data augmentation requires 700 hours to achieve

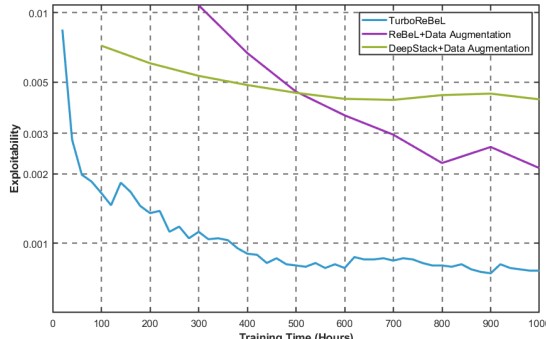

Figure 2: Exploitability of ReBeL, DeepStack (both with $K = 24$ isomorphic data augmentation) and TurboReBeL (with $K = 2$ isomorphic data augmentation) over training time on the same server for TEH game.

the same level. These results indicate that TurboReBeL not only trains substantially faster than ReBeL or DeepStack but also achieves lower final exploitability. According to ReBeL's experimental results on TEH (Brown et al., 2020), achieving an exploitability of 0.002 requires 51 million samples, whereas TurboReBeL achieves the same level of exploitability with only approximately $150,000$ samples. These results indicate that TurboReBeL attains equivalent exploitability with a training time (cost) reduction of approximately $250\times$ compared to ReBeL.

We further evaluated heads-up performance on HUNL against Slumbot (Jackson, 2013), the champion of the Annual Computer Poker Competition (Bard et al., 2013). The evaluation included over $500,000$ hands using advanced variance reduction techniques (Burch et al., 2018). As summarized in Table 1, TurboReBeL achieves a win-rate of $42 \pm 12$ mbb/hand using only 10 million raw samples. In contrast, ReBeL requires 4.5 billion samples to achieve a comparable win-rate of $45 \pm 5$ mbb/hand. This represents a 450-fold reduction in sample requirements compared to ReBeL. Even after considering for the additional computational overhead from data generation (Phase 2 of Algorithm 2) and network training (Appendix B), TurboReBeL maintains $\sim 250\times$ overall improvement in training efficiency while matching the empirical performance of ReBeL. This gain is particularly significant given that the enormous state space of HUNL (over $10^{161}$ information sets) (Johanson, 2013), which traditionally requires prohibitive computational resources.

Table 1: Sample efficiency and win-rates (measured by mbb/hand) against Slumbot.

| Method | Samples | Win-rate |
|---|---|---|
| ReBeL (Brown et al., 2020) | 4.5 billion | $45 \pm 5$ |
| SoG (Schmid et al., 2023) | 1.1 billion | $7 \pm 3$ |
| TurboReBeL (**Ours**) | 10 million | $42 \pm 12$ |

In summary, our experiments on both TEH and HUNL demonstrate that TurboReBeL achieves $\sim 250\times$ training acceleration while maintaining performance comparable to ReBeL. This successfully addresses the fundamental efficiency bottlenecks that have limited previous depth-limited solvers. By integrating single-sample multi-iteration generation and isomorphic data augmentation, TurboReBeL significantly reduces training costs while preserving strong scalability, low exploitability, convergence guarantees, human-data-free training, and practical real-time decision capability. This advancement not only enhances the practicality of depth-limited solving in large IIEFGs but also opens new avenues for research in efficient equilibrium-finding algorithms.

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

# A    DESCRIPTION OF GAMES

This section provides a formal description of the two large IIEFGs used for evaluation: Heads-Up No-Limit Texas Hold'em (HUNL) and Turn Endgame Hold'em (TEH). The games and settings are same as used in ReBeL. The game rules and evaluation settings are identical to those used in the original ReBeL implementation (Brown et al., 2020).

HUNL is the two-player, zero-sum variant of no-limit Texas Hold'em. The game uses a standard 52-card deck with 4 suits and 13 ranks. The game protocol is as follows:

- *Private cards:* Each player receives 2 private cards that are visible only to themselves.
- *Betting rounds:* The game proceeds through up to 4 betting stages:
    1. *Pre-flop:* Players are assigned to the small blind (SB) and big blind (BB) positions. The SB player posts 50 chips and the BB player posts 100 chips to form the initial pot. SB player acts first.
    2. *Flop:* three public cards are dealt, and then BB player acts first.
    3. *Turn:* one additional public card is dealt, and then BB player acts first.
    4. *River:* one final public card is dealt, and then BB player acts first.
- *Action space:* At each decision point, players can fold, check/call, or raise. Raises must be at least the size of the big blind and at least equal to any previous raise in the same round. The maximum raise is limited by a player's remaining chip stack (all-in).
- *Termination:* The hand ends when either: (i) a player folds (opponent wins the pot); or (ii) all stages complete and players reveal their private cards to determine the best 5-card hand using any combination of private and public cards.
- *Chip stacks:* During training, initial stacks vary randomly between 1 to 250 big blinds and we allow non-integer bets. During testing, both players start with 200 big blinds and 20,000 chips, and all bet chips are integer.

TEH is a simplified variant of HUNL designed for efficient exploitability measurement. Both players automatically check/call through pre-flop and flop stages, beginning actual decision-making at the turn stage. During training, initial stacks vary between 1 to 50 big blinds. For exploitability testing, both players start with 20 big blinds and 2,000 chips. For standardized evaluation, the first four community cards are fixed as 3♠7♡10♢K♠ during exploitability measurement. This configuration maintains strategic complexity while enabling tractable exploitability computation through state space reduction.

The competitive performance is measured in milli-big-blinds per hand (mbb/hand), where 1 mbb/-hand = 0.001 big blinds per hand. For example, a win rate of 10 mbb/hand indicates an average profit of 0.01 big blinds per hand. During testing, players exchange positions after each hand.

# B    SETTINGS

This section details the neural network architecture, hyperparameters, and computational resources used in our TurboReBeL implementation to ensure reproducibility and facilitate future research in large IIEFGs. Our setup closely follows that of ReBeL (Brown et al., 2020) with modifications to accommodate our accelerated training framework.

## B.1    NEURAL NETWORK ARCHITECTURE

We use a 6-layer multilayer perceptron with the following specifications:

- *Input representation:* The input features include 2,780 dimension:
    - Public cards: 52-dimensional one-hot encoding + 13-dimensional rank encoding + 4-dimensional suit encoding + 5-dimensional flop/turn/river stage encoding
    - Chip information: 4 features (each player's current pot contribution and remaining chips relative to the big blind)

- Current player: 2-dimensional one-hot encoding
- Action history: 48-dimensional vector, representing history action sequence
- Range distribution: $2,652$-dimensional vector (Normalized probability of each possible hand)

- *Network structure:*
  - 6 fully connected layers with hidden dimension of $1,536$
  - ReLU activation functions with Layer Normalization
- *Output representation:* $2,652$-dimension expected value vector (relative to current pot size)
- *Loss function:* Huber loss

### B.2 TRAINING CONFIGURATION

- *Exploration:* $25\%$ probability of taking random actions during self-play
- *Replay buffer size:* 120 million
- *Sampling:* Each epoch gains $30,000$ raw samples
- *Optimizer:* Adam
- *Learning rate:* 0.0003, halved every 25 epochs
- *Batch size:* $1,024$
- *Isomorphic Data Augmentation:* $K = 2$ transformations

### B.3 COMPUTATIONAL RESOURCES

- *GPU:* NVIDIA A100 80GB PCIe GPU
- *CPU:* Intel(R) Xeon(R) Gold 6348 2.60GHz CPU
- *Sample generation:* 3 GPUs + 96 CPUs
- *Network training:* 1 dedicated GPU

### B.4 GAME TREE

- *Limited Lookahead:* Depth-limited subgames are built until the current stage ends or the next stage begins
- *Betting abstraction:*
  - First two levels: raise sizes of $0.5\times$, $1\times$, and $2\times$ pot, plus all-in
  - Remaining levels: raise sizes of $0.75\times$ pot plus all-in
  - Raise sizes are randomly scaled by $[0.75, 1.25]$ during training
- *CFR*: Discounted CFR (Brown & Sandholm, 2019b) with $\alpha = 1.5, \beta = 0, \gamma = 2$ and $T = 250$ iterations.

## C LLM USAGE

Large language models are used to polish sentences only. Our idea and code implementation are completely independent of large language models.

## D PROOFS

In this appendix, we present the detailed proof for Theorems 1 and 2. Our proofs are based on several works (Hart & Mas-Colell, 2000; Zinkevich et al., 2007; Burch et al., 2014; Moravcik et al., 2016; Brown et al., 2020). We will make use of the following notation.

**Notation.** Define the composite strategy

$$\overline{\sigma}^{t,T} := \begin{cases} \overline{\sigma}^t & \text{(inside the depth-limited subgame)}, \\ \overline{\sigma}^T & \text{(everywhere else)}. \end{cases}$$

### D.1 Proof of Theorem 1

**Theorem 1 (Value Estimation Consistency).** Let $T$ be the total CFR iteration number of TurboReBeL and $\overline{\sigma}^T$ be the reference strategy for the full game. During the depth-limited subgame solving period in the TurboReBeL, let $\overline{\sigma}^t$ be the average strategy in this depth-limited subgame at iteration $t$ of CFR ($t \leq T$). Let $s$ be an arbitrary leaf node in the depth-limited subgame, and let $\beta_{s,t}$ be any leaf PBS reached according to $\overline{\sigma}^t$. Let $\sigma^*$ be a Nash equilibrium strategy for full game. The value network output for any information set $I_p \in s$ is $\hat{v}_p^\theta(I_p \mid \beta_{s,t})$ and we set the maximum estimation error of the neural network as $\varepsilon_{\text{approx}} := \sup_{I_p \in \beta_{s,t}} \left| \hat{v}_p^\theta(I_p \mid \beta_{s,t}) - v_p^{\overline{\sigma}^T}(I_p \mid \beta_{s,t}) \right|$. During depth-limited subgame solving at iteration $t$, the value estimation error for any information set $I_p \in s$ satisfies:

$$\left| v_p^\theta(I_p \mid \beta_{s,t}) - v_p^{\sigma^*}(I_p) \right| \leq \varepsilon_{\text{approx}} + O\left(1/\sqrt{t}\right).$$

*Proof.* Start with the triangle inequality

$$\left| \hat{v}_p^\theta(I_p \mid \beta_{s,t}) - v_p^{\sigma^*}(I_p) \right| \leq \underbrace{\left| \hat{v}_p^\theta(I_p \mid \beta_{s,t}) - v_p^{\overline{\sigma}^{t,T}}(I_p) \right|}_{\Delta(\theta)} + \underbrace{\left| v_p^{\overline{\sigma}^{t,T}}(I_p) - v_p^{\sigma^*}(I_p) \right|}_{\Delta(I_p)}.$$

**(A) Bounding $\Delta(\theta)$.** Note that $\beta_{s,t}$ is generated via strategy $\overline{\sigma}^{t,T}$, and the value network data is generated via strategy $\overline{\sigma}^T$. Both $\overline{\sigma}^{t,T}$ and $\overline{\sigma}^T$ adopt $\overline{\sigma}^T$ after $s$, so $v_p^{\overline{\sigma}^{t,T}}(I_p) = v_p^{\overline{\sigma}^T}(I_p \mid \beta_{s,t})$. Thus, $\Delta(\theta)$ does not exceed $\varepsilon_{\text{approx}}$.

**(B) Decomposing $\Delta(I_p)$.** We decompose $\Delta(I_p)$ by inserting the reference strategy $\overline{\sigma}^T$ as an intermediate strategy profile:

$$\Delta(I_p) \leq \left| v_p^{\overline{\sigma}^{t,T}}(I_p) - v_p^{\overline{\sigma}^T}(I_p) \right| + \left| v_p^{\overline{\sigma}^T}(I_p) - v_p^{\sigma^*}(I_p) \right|. \tag{1}$$

Next, we bound the two terms in equation 1 separately.

**(C) Bounding $\left| v_p^{\overline{\sigma}^T}(I_p) - v_p^{\sigma^*}(I_p) \right|$.** The averaged-CFR convergence guarantee implies that the continuation values computed under $\overline{\sigma}^T$ uniformly approximate those under $\sigma^*$ at rate $O\left(1/\sqrt{T}\right)$. Thus, for all information set $I_p$, we have

$$\left| v_p^{\overline{\sigma}^T}(I_p) - v_p^{\sigma^*}(I_p) \right| \leq O\left(\frac{1}{\sqrt{T}}\right).$$

**(D) Bounding $\left| v_p^{\overline{\sigma}^{t,T}}(I_p) - v_p^{\overline{\sigma}^T}(I_p) \right|$.** These two profiles differ only inside the depth-limited subgame (before $s$): $\overline{\sigma}^{t,T}$ uses $\overline{\sigma}^t$ in the depth-limited subgame while $\overline{\sigma}^T$ uses the reference strategy. For any continuation history $h$ at or after $s$, the payoff $u_p(h)$ depends only on the same reference strategy $\overline{\sigma}^T$. Thus, for any history $h \in s$, we have $u_p^{\overline{\sigma}^{t,T}}(h) = u_p^{\overline{\sigma}^T}(h)$.

Because $\overline{\sigma}^T$ is fixed, we can obtain the perfect value of the leaf history, which is fixed during CFR this depth-limited subgame. Let $\overline{\sigma}^{*,T}$ be the optimal strategy in a depth-limited subgame with strategy $\overline{\sigma}^T$ outside the known depth-limited subgame. For any $I_p \in s$, we have:

$$\left| v_p^{\overline{\sigma}^{t,T}}(I_p) - v_p^{\overline{\sigma}^T}(I_p) \right| \leq \left| v_p^{\overline{\sigma}^{t,T}}(I_p) - v_p^{\overline{\sigma}^{*,T}}(I_p) \right| + \left| v_p^{\overline{\sigma}^T}(I_p) - v_p^{\overline{\sigma}^{*,T}}(I_p) \right| \leq O\left(\frac{1}{\sqrt{t}}\right) + O\left(\frac{1}{\sqrt{T}}\right)$$

This bound holds because both strategies use the same reference strategy $\overline{\sigma}^T$ outside the depth-limited subgame, and the internal strategy difference is controlled by CFR's convergence rate.

**(E) Combining all terms.** Under the assumption $t \leq T$, and from (1) we obtain

$$\Delta(I_p) \leq O\left(\frac{1}{\sqrt{t}}\right).$$

Substituting back and combining with the network fit term yields the stated bound

$$\left| \hat{v}_p^\theta(I_p \mid \beta_{s,t}) - v_p^{\sigma^*}(I_p) \right| \leq \varepsilon_{\text{approx}} + O\left(\frac{1}{\sqrt{t}}\right).$$

This completes the proof. $\square$

## D.2 PROOF OF THEOREM 2

**Theorem 2 (Convergence of TurboReBeL).** Assume that, for every depth-limited subgame, the PBS value network provides CFV estimates whose error, for any information set used in iteration $t \leq T$, satisfies $\left|\hat{v}_p^\theta(I_p) - v_p^{\sigma^*}(I_p)\right| \leq \varepsilon_{\text{approx}} + O\left(\frac{1}{\sqrt{t}}\right)$. Then the exploitability of the strategy that applies $T$-iteration CFR subgame solving in TurboReBeL is bounded by

$$\text{Expl}(\overline{\sigma}^T) = O\left(\varepsilon_{\text{approx}} + \frac{1}{\sqrt{T}}\right).$$

*Proof.* **(A) From leaf CFV bias to per-iteration regret bias.** By Theorem 1, every leaf CFV target used at iteration $t$ deviates from the equilibrium continuation value by at most

$$\delta_t := \sup_{I_p \text{ is a leaf}} \left|\hat{v}_p^\theta(I_p) - v_p^{\sigma^*}(I_p)\right| \leq \varepsilon_{\text{approx}} + O(1/\sqrt{t}).$$

A CFR update depends only on differences of continuation values. Perturbing each continuation by at most $O(\delta_t)$ perturbs every action-value difference by at most $O(\delta_t)$. Thus, each instantaneous regret increment at any information set on iteration $t$ is changed by at most $O(\delta_t)$.

**(B) Accumulating bias across iterations.** Denote $R_p^T(I)$ the cumulative regret at information set $I$ after $T$ biased CFR iterations. We can decompose it into the following

$$R_p^T(I) \leq R_p^{T,\text{CFR}}(I) + R_p^{T,\text{bias}}(I),$$

where $R_p^{T,\text{CFR}}(I)$ is the regret under exact CFR (oracle leaves), and $R_p^{T,\text{bias}}(I)$ is the additional regret caused by biased leaf targets.

The exact CFR term guarantees

$$R_p^{T,\text{CFR}}(I) \leq O(\sqrt{T}).$$

Meanwhile, the bias term satisfies

$$R_p^{T,\text{bias}}(I) \leq \sum_{t=1}^T O(\delta_t).$$

Hence, for all information sets $I$, we have

$$\frac{R_p^T(I)}{T} \leq O\left(\frac{1}{\sqrt{T}}\right) + \frac{1}{T}\sum_{t=1}^T O(\delta_t).$$

Substituting $\delta_t \leq \varepsilon_{\text{approx}} + O(1/\sqrt{t})$, and using $\sum_{t=1}^T 1/\sqrt{t} \leq 2\sqrt{T}$, we obtain that

$$\frac{1}{T}\sum_{t=1}^T O(\delta_t) = O(\varepsilon_{\text{approx}}) + \frac{1}{T}\sum_{t=1}^T O\left(\frac{1}{\sqrt{t}}\right) = O(\varepsilon_{\text{approx}}) + O\left(\frac{1}{\sqrt{T}}\right).$$

Thus, we have that for every information set $I$,

$$\frac{R_p^T(I)}{T} \leq O\left(\varepsilon_{\text{approx}} + \frac{1}{\sqrt{T}}\right). \tag{2}$$

**(C) From average regret to exploitability.** The exploitability of the averaged strategy $\overline{\sigma}^T$ in the subgame is bounded above by the sum average regret across all information sets. Thus, by applying (2), we obtain

$$\text{Expl}_{\text{sub}}(\overline{\sigma}^T) \leq O\left(\varepsilon_{\text{approx}} + \frac{1}{\sqrt{T}}\right). \tag{3}$$

**(D) Subgame solving safety.** When the public state $s$ is reached, before entering the depth-limited subgame rooted at $s$, we construct a re-solving game in which each player $p$ at the root information set $I_p \in s$ receives a gift value $g_p(I_p) := v_p^{\overline{\sigma}^T}(I_p)$. Since the gift values are defined using $\overline{\sigma}^T$, whose value error is already bounded by Theorem 1, all resulting gift inequalities hold up to the same $O\left(\varepsilon_{\text{approx}} + \frac{1}{\sqrt{T}}\right)$ error.

In the re-solving game, player $p$ may choose to "exit" immediately and receive $g_p(I_p)$, or enter the depth-limited subgame and follow a locally re-solved strategy. After running $T$ iterations of local CFR inside the subgame using TurboReBeL, the returned strategy $\sigma^{\text{re}}$ satisfies

$$v_p^{\sigma^{\text{re}}}(I_p) \geq g_p(I_p) - O\left(\frac{1}{\sqrt{T}} + \varepsilon_{\text{approx}}\right), \qquad I_p \in s. \tag{4}$$

Inequality (4) implies that the opponent cannot exploit the re-solved strategy by more than the same order of error, since giving the player a guaranteed fallback value prevents any additional loss. Specifically, for the opponent $p$,

$$v_p^{(\sigma_{-p}^{\text{re}}, \text{BR}(\sigma_{-p}^{\text{re}}))}(I_p) \leq v_p^{(\overline{\sigma}_{-p}^T, \text{BR}(\overline{\sigma}_{-p}^T))}(I_p) + O\left(\frac{1}{\sqrt{T}} + \varepsilon_{\text{approx}}\right), \qquad I_p \in s. \tag{5}$$

Combining (5) with the depth-limited subgame exploitability bound 3, the exploitability of the re-solved strategy in the subgame rooted at $s$ satisfies

$$\text{Expl}_{\text{sub}}(\sigma^{\text{re}}) = O\left(\varepsilon_{\text{approx}} + \frac{1}{\sqrt{T}}\right).$$

**(E) Full-game exploitability.** The full game can be viewed as a finite stack of depth-limited subgames. Since each subgame is solved safely with error $O(\varepsilon_{\text{approx}} + 1/\sqrt{T})$, and the number of such levels is a game-dependent constant, the global strategy $\overline{\sigma}^T$ produced by TurboReBeL with re-solving inherits the same order guarantee:

$$\text{Expl}(\overline{\sigma}^T) = O\left(\varepsilon_{\text{approx}} + \frac{1}{\sqrt{T}}\right).$$

This completes the proof.

$\square$

# E    Implementation of TurboReBeL

We provide the key implementation details for TurboReBeL. This implementation could be combined with the open-source ReBeL framework, ensures the reproducibility of our results. Our system is implemented by ourselves in C++ with parallel computing support for efficient intermediate value computation. The leaf node value estimation is performed using PyTorch's C++ (LibTorch), which allows seamless integration of pre-trained neural networks into the game-solving pipeline. To maximize GPU utilization, we aggregate all leaf nodes into a single batch before inference, significantly reducing overhead from repeated kernel launches.

```cpp
// Leaf payoff estimation via neural network
void estimate_leaf_payoffs(Game_Tree& tree, vector<Node> leaf_nodes) {
    vector<float> network_input = tree.encoding(leaf_nodes);
    if(network_input.empty())return;
    // Transfer input to GPU tensor
    torch::DeviceType device = at::kCUDA;
    torch::Tensor state_tensor = at::from_blob(
        network_input.data(),
        {(int)network_input.size() / INPUT_DIM, INPUT_DIM},
        at::TensorOptions().dtype(at::kFloat)
    ).to(device);
```

```
12    std::vector<torch::jit::IValue>inputs{state_tensor};
13    auto output = leaf_value_network.forward(inputs).toTensor();
14    at::Tensor leaf_estimations = output.cpu().detach();
15    tree.assign_leaf_payoffs(leaf_nodes, leaf_estimations);
16 }
17 //TurboReBeL Sampling Function
18 void sampling(Node root, PBS average_pbs, vector<PBS> intermediate_pbs) {
19    Strategy strategy_profile[T + 1], average_strategy_profile[T + 1];
20    Game_Tree tree = depth_limited_game_tree(root);
21    vector<Node> leaf_nodes = tree.get_leaf_nodes();
22    vector<Node> terminal_nodes = tree.get_terminal_nodes();
23    // Phase 1: Strategy Evolution via CFR
24    strategy_profile[0] = tree.uniform_strategy();
25    average_strategy_profile[0] = tree.uniform_strategy();
26    for(int t = 1; t <= T; t++) {
27        tree.update_reach_prob(average_pbs, average_strategy_profile[t -
              ↪ 1]);
28        estimate_leaf_payoffs(tree, leaf_nodes);
29        tree.update_reach_prob(average_pbs, strategy_profile[t - 1]);
30        tree.update_terminal_payoffs(terminal_nodes);
31        tree.update_cfv(strategy_profile[t - 1]);
32        tree.update_regret(t);
33        strategy_profile[t] = tree.update_policy(t);
34        average_strategy_profile[t] = update_average_policy(
              ↪ average_strategy_profile[t - 1], strategy_profile[t], t);
35    }
36    Strategy reference_strategy = strategy_profile[T];
37    // Phase 2: Multi-Iteration Data Generation
38    for(int t = 0; t <= T; t++) {
39        tree.update_reach_prob(intermediate_pbs[t], reference_strategy);
40        estimate_leaf_payoffs(tree, leaf_nodes);
41        tree.update_terminal_payoffs(terminal_nodes);
42        vector<float> root_value = tree.update_cfv(reference_strategy);
43        add_data_with_isomorphic(intermediate_pbs[t], root_value);
44    }
45    Node leaf = sample_leaf(root, reference_strategy);
46    if(!is_terminal_node(leaf)) {
47        average_pbs = get_belief(average_pbs, reference_strategy, leaf);
48        intermediate_pbs.clear();
49        for(int t = 0; t <= T; t++)
50            intermediate_pbs.push_back(get_belief(average_pbs,
                  ↪ average_strategy_profile[t], leaf));
51        sampling(leaf, average_pbs, intermediate_pbs);
52    }
53 }
```

Listing 1: TurboReBeL Sampling Procedure with Batch Leaf Evaluation

## F CONCLUSION

TurboReBeL establishes a new state-of-the-art for scalable depth-limited solving in IIEFGs by fundamentally rethinking the data generation process. Its core innovations, including Single-Sample Multi-Iteration Generation and Isomorphic Data Augmentation, resolve the critical sampling efficiency bottlenecks that have long constrained previous approaches. The result is a paradigm shift: achieving a $\sim250\times$ reduction in training cost without compromising the strong scalability, low exploitability, theoretical guarantees, or fast real-time search. By making large IIEFG training computationally practical, TurboReBeL opens the door to exploring increasingly complex strategic settings and reinforces the power of combining search, learning, and game-theoretic principles. We believe this work paves the way for a new generation of multi-agent AI systems capable of sophisticated strategic reasoning in the complex environments.

