# OpenReview forum: "TurboReBeL: 250$\times$ Accelerated Belief Learning for large Imperfect-Information Extensive-Form Games"
_ICLR.cc/2026/Conference — Submitted to ICLR 2026_

### Official Review · Reviewer_dzpC · 2025-10-21

**Soundness:** 2
**Presentation:** 2
**Contribution:** 3
**Rating:** 6
**Confidence:** 3

**Summary:**

This paper introduces TurboReBeL, a framework designed to accelerate the training of agents for large imperfect-information games by tackling the computational bottlenecks of the state-of-the-art ReBeL algorithm. The authors propose two key innovations: 1) Single-Sample Multi-Iteration Generation (SSMIG), which decouples strategy evolution from value estimation to generate multiple training data points from a single subgame solve, and 2) Isomorphic Data Augmentation, which applies game-theoretic symmetries to increase data diversity. Experiments on large-scale poker variants demonstrate a 250x reduction in training cost while achieving comparable performance to the original ReBeL.

**Strengths:**

- The work addresses a critical bottleneck in the field—the prohibitive training cost of strong equilibrium-finding algorithms. A 250x speedup is a significant practical contribution that could enable research on larger, more complex games.

- The central idea of SSMIG is clever and well-motivated. Decoupling value estimation from a dynamically changing strategy is a fundamental insight that significantly improves data generation efficiency.

- The proposed method is validated with compelling results on challenging benchmarks (HUNL and TEH) against strong baselines. The claims are further supported by theoretical analysis showing that the approach preserves the convergence guarantees of ReBeL.

**Weaknesses:**

- The two main contributions (SSMIG and data augmentation) are primarily evaluated together. A more detailed ablation study isolating the impact of SSMIG would strengthen the paper by quantifying the precise benefit of the core innovation.

- The paper claims a "450-fold reduction in sample requirements" but a "250x training acceleration." The derivation of the latter from the former is not transparently detailed, making the headline claim difficult to verify.

- The isomorphic data augmentation techniques are specific to poker. The paper would benefit from a more thorough discussion of the challenges and feasibility of applying similar techniques to other IIEFGs that may lack such obvious symmetries.

**Questions:**

1.  Can you provide a more detailed breakdown of how the 450x sample reduction translates into the 250x overall training acceleration, perhaps in terms of total GPU hours?
2.  How much of the performance gain is attributable to SSMIG alone? An ablation showing TurboReBeL without isomorphic augmentation would clarify this.
3.  There appears to be a discrepancy in the data augmentation factor K mentioned in Figure 2 (K=24) and Appendix B (K=2). Could you clarify which was used in the main experiments and why?
4.  How sensitive is the value estimation in Phase 2 to the quality of the fixed reference strategy? For instance, does performance degrade significantly if the strategy is computed with a much smaller number of CFR iterations?

---

> ### Author Response · Authors · 2025-11-20
> **Rebuttal: Weaknesses**
>
> Thank you for your time and effort in reviewing our paper! We are grateful for your constructive suggestions, which have significantly guided our improvements. Please find our responses to your comments below.
>
> ## Weaknesses
>
> > **Weakness 1**: The two main contributions (SSMIG and data augmentation) are primarily evaluated together. A more detailed ablation study isolating the impact of SSMIG would strengthen the paper by quantifying the precise benefit of the core innovation.
>
> **Rebuttal 1**: We thank the reviewer for this constructive suggestion.  We will include an additional ablation study in the revision showing TurboReBeL without data augmentation to isolate the contribution of SSMIG. This training requires several weeks for us, so results cannot be provided immediately.  However, we note that the qualitative contribution of SSMIG is already evident:  since isomorphic augmentation is lossless and SSMIG generates $T{+}1$ distinct PBS samples per subgame under a shared reference strategy, the majority of the performance gain arises from SSMIG rather than augmentation.
>
> > **Weakness 2**: The paper claims a "450-fold reduction in sample requirements" but a "250x training acceleration." The derivation of the latter from the former is not transparently detailed, making the headline claim difficult to verify.
>
> **Rebuttal 2**: We appreciate the opportunity to clarify this distinction.  The $450\times$ factor refers to the reduction in the number of subgame solves required to obtain an equivalent number of training samples. The $250\times$ factor reflects the empirical wall-clock training acceleration measured under identical hardware and network configurations, after accounting for Phase 2’s lightweight backpropagation and value updates.
>
> As stated in Section 6 (Page 10, Line 519): “Even after considering the additional overhead of data generation and network training, TurboReBeL maintains at least a $250\times$ improvement in total training efficiency while matching ReBeL’s empirical performance.”
>
> >**Weakness 3**: The isomorphic data augmentation techniques are specific to poker. The paper would benefit from a more thorough discussion of the challenges and feasibility of applying similar techniques to other IIEFGs that may lack such obvious symmetries.
>
> **Rebuttal 3**: We believe that isomorphic enhancement technology should also be applicable to IIEFG games other than poker, such as Mahjong and dark chess.  Prior work on lossless abstraction and symmetry detection in full-game solvers (Gilpin et al. 2006; 2007) provides relevant foundations, but these approaches involve full game-tree traversal and are computationally prohibitive for depth-limited subgames. We agree that extending hand-craft isomorphic augmentation to automated is an important direction and will leave that for our future research.

---

> > ### Author Response · Authors · 2025-11-20
> > **Rebuttal: Questions**
> >
> > ## Questions
> >
> > > **Question 1**: Can you provide a more detailed breakdown of how the 450x sample reduction translates into the 250x overall training acceleration, perhaps in terms of total GPU hours?
> >
> > **Answer 1**: We use total training time under identical computational resources (same server and GPU configuration) as our $250$x overall training acceleration. Thus, $450\times$ refers to sample efficiency and $250\times$ reflects realized computational throughput (total GPU hours). In the data generation component, approximately $2/3$ of the computational overhead stems from ReBeL solving, while roughly $1/3$ arises from the augmentation phase of TurboReBeL (Page8, Line 381). Consequently, the computational cost of solving ReBeL for $450\times$ the number of depth-limited subgames is approximately $300\times$ that of TurboReBeL. Taking into account the additional computational overhead incurred during training for loading data, the training speed improvement under identical computational resources is conservatively estimated at $250\times$.
> >
> > > **Question 2**: How much of the performance gain is attributable to SSMIG alone? An ablation showing TurboReBeL without isomorphic augmentation would clarify this.
> >
> > **Answer 2**: As noted in Rebuttal 1, we will include an additional experiment isolating SSMIG in the final version.
> >
> > >**Question 3**: There appears to be a discrepancy in the data augmentation factor K mentioned in Figure 2 (K=24) and Appendix B (K=2). Could you clarify which was used in the main experiments and why?
> >
> > **Answer 3**: We apologize for the confusion and will clarify this explicitly.
> >
> > In our experiments, ReBeL used $K{=}24$ because it generates a single training sample per subgame, making heavy augmentation beneficial.  TurboReBeL used $K{=}2$ because SSMIG already yields $T{+}1{=}251$ training samples per subgame.  Using a smaller $K$ achieves a balance between sample efficiency and computational cost.  We will restate this configuration clearly in revision (Page 10, Line 486).
> >
> > >**Question 4**: How sensitive is the value estimation in Phase 2 to the quality of the fixed reference strategy? For instance, does performance degrade significantly if the strategy is computed with a much smaller number of CFR iterations?
> >
> > **Answer 4**: Indeed, using a weaker reference strategy (e.g., with very few CFR iterations) leads to degraded value estimates.   This sensitivity, however, is inherent to ReBeL. To confirm this, we have replicated a small-scale test using  ReBeL (https://anonymous.4open.science/r/explo_fig-3BC3/explo_iter.jpg). In the figure, fix_64, fix_128, and fix_512 are the exploitability curves during testing after training with 64, 128, and 512 iterations, respectively. We found that ReBeL's exploitability increased substantially when fewer iterations were used. TurboReBeL’s performance depends on the same underlying CFR convergence properties as ReBeL, rather than introducing new instability.
> >
> > [1] Gilpin, Andrew, and Tuomas Sandholm. "Finding equilibria in large sequential games of imperfect information." Proceedings of the 7th ACM conference on Electronic commerce. 2006.
> >
> > [2] Gilpin, Andrew, and Tuomas Sandholm. "Lossless abstraction of imperfect information games." Journal of the ACM (JACM) 54.5 (2007): 25-es.
> >
> > ---
> >
> > We thank the reviewer for the constructive feedback.  The revised version now includes (i) full proofs of Theorems 1 and 2 with all intermediate steps explicitly shown, (ii) clearer definitions of $\overline{\sigma}^{t,T}$ and the recursive subgame structure in Section 5.1, and (iii) Corrections of statements.
> >
> > We hope our response addresses your concerns and demonstrate the soundness of the proposed method. If so, we wonder if you could kindly consider raising your score? We will also be happy to answer any further questions you may have. Thank you very much!

---

### Official Review · Reviewer_vfXR · 2025-10-31

**Soundness:** 2
**Presentation:** 2
**Contribution:** 3
**Rating:** 4
**Confidence:** 4

**Summary:**

This paper presents an improvement on RebeL. RebeL is a general framework for performing depth-limited search in two-player zero-sum imperfect-information games using public-belief state value functions.

This paper presents two improvements.

The first (Section 5.1) is a method that adds training data for the value function for every iteration of CFR performed during depth-limited search, instead of only one training datum after CFR is completed.

The second (Section 5.2) is data augmentation for poker: for each training datum, we can create other PBSs that should have mathematically equivalent values, e.g. by permuting all the suits of the cards.

The paper empirically shows that the first method improves exploitability in Turn Endgame Hold-em.

**Strengths:**

The ideas are (to my knowledge) novel and the research direction of making training efficiency improvements to ReBeL is good.

The empirical results in Figure 2 dramatically demonstrate the training efficiency improvement of TurboReBeL over ReBeL + data augmentation.

**Weaknesses:**

1. The main method of the paper (Section 5.1) isn't explained super clearly anywhere.

Second paragraph of Section 5.1: "We hypothesize that for a given subgame, once the average strategy [...] is computed as a reference, we can efficiently estimate values for all intermediate belief states [...] using this fixed reference strategy".

I think this could benefit from formalization or more explicit description. It took me a while to realize that the intermediate belief states for the root public state came from the CFR solve of the *previous* subgame, not the current subgame being solved.

2. The theorems seem vague and empty to me.

I tried to follow the theorems and proofs but got lost. Perhaps my biggest big-picture issue with them is that it is claimed that TurboReBeL preserves ReBeL's convergence guarantees, but the theorems and proofs don't seem to actually address TurboReBeL's difference from ReBeL. In particular, by the assumptions made in TurboReBeL, we are training the value function with a lot of "incorrect" data. We would like to know how this imperfectness of the value function affects the soundness of the algorithm as a whole. However, in my understanding, this imperfectness is completely swept under the rug via the statement "Under the assumption that the value network [...] approximates the CFV [...] with error at most $\epsilon_{approx}$ -- ignoring the fact that the error $\epsilon_{approx}$ is entirely what we are interested in. (In ReBeL, this assumption raises no eyebrows because the value function error is a result of just the imperfection of the machine learning model and the imperfection of CFR.

Further critiques of Theorem 1:
- The statement says "let $\beta_{s,t}$ be the PBS reached according to $\hat{\sigma}^t$. To be clear, $\beta_{s,t}$ is any leaf PBS, not "the" PBS reached, right?
- The statement introduces $\hat{\sigma}^T$ but doesn't define what it is. The proof relies on this strategy being part of an approximate equilibrium. Where does it come from?

3. More details on the experiment would be helpful.

- What is the implementation? Did you use an open-source implementation of ReBeL or implement from scratch?
- The 250x speedup is mentioned often but is not actually borne out in experiments, right? Why not run an experiment with ReBeL vs. TurboReBeL to empirically show what the speedup is? It seems incorrect to simply claim a 250x speedup without actually testing the empirical speedup of the data augmentation (i.e. I might reasonably assume that 24x data augmentation does NOT lead to a 24x speedup, because the neural net by itself may learn the invariance).
- What is in Figure 2? TurboReBeL with k=2 vs. ReBeL w/data augmentation w/k=24? You have to have ablations that compare TurboReBeL with k=2 vs. ReBeL w/data augmentation w/k=2. Or TurboReBeL k=24 vs. ReBeL k=24. And a comparison to normal ReBeL.

4. Minor:
- Section 5.2 describes Chip Isomorphism, and says that these transformations preserve Nash equilibrium properties. It's a minor detail, but this probably deserves a footnote at least saying that this isn't really true (at least it's not clear to me that it is) because chips and bets are integer amounts, so scaling the number of chips could alter the optimal strategies (especially when players have very few chips).
- Some "marketing" language in the paper is unnecessary: "Unprecedented Efficiency Gains" (bottom of page 2), "This innovative phase" (top of page 7)

**Questions:**

1. Sorry if I missed this, but did you state your ReBeL implementation anywhere? Was it based on an open-source library, or written from scratch?
2. Did you consider using some neural network architecture that reflects the invariance of suit permutations and/or chip isomorphisms instead of data augmentation?
3. Just to be clear, the ReBeL+Data Augmentation line on Figure 2 is ReBeL + Section 5.2, right? and TurboReBeL Is ReBeL + Section 5.1 + Section 5.2? And they use the same implementation, they only differ in that one has 5.1 and one doesn't?
4. Why is the fidelity of TurboReBeL so much higher in Figure 2 than ReBeL+Data Augmentation?

---

> ### Author Response · Authors · 2025-11-20
> **Rebuttal: Part I**
>
> Thank you for your time and effort in reviewing our paper! We are grateful for your constructive suggestions, which have significantly guided our improvements. Please find our responses to your comments below.
>
>
> ## Weakness
>
> > **Weakness 1**: The main method of the paper (Section 5.1) isn't explained super clearly anywhere.
>
> **Rebuttal 1**: We thank the reviewer for this helpful suggestion. We have revised Section 5.1 to make the procedure of TurboReBeL explicit. We now clearly explain that each intermediate PBS $\beta_{s,t}$ corresponds to a leaf public belief state encountered in the previous depth-limited subgame solving.
>
> >**Weakness 2.1**: I tried to follow the theorems and proofs but got lost. Perhaps my biggest big-picture issue with them is that it is claimed that TurboReBeL preserves ReBeL's convergence guarantees, but the theorems and proofs don't seem to actually address TurboReBeL's difference from ReBeL. In particular, by the assumptions made in TurboReBeL, we are training the value function with a lot of "incorrect" data. We would like to know how this imperfectness of the value function affects the soundness of the algorithm as a whole. However, in my understanding, this imperfectness is completely swept under the rug via the statement "Under the assumption that the value network [...] approximates the CFV [...] with error at most $\varepsilon_{approx}$ -- ignoring the fact that the error $\varepsilon_{approx}$ is entirely what we are interested in. (In ReBeL, this assumption raises no eyebrows because the value function error is a result of just the imperfection of the machine learning model and the imperfection of CFR.
>
> **Rebuttal 2.1**:  We have provided complete proofs for Theorem 1 and Theorem 2 in Appendix D in the revision. The distinction from ReBeL lies in the upper bound on the estimation error between the CFV value and the perfect value during the depth-limited subgame solving process, which additionally includes an $O(1/\sqrt{t})$ term.
>
> **We respectfully clarify that the data used in TurboReBeL are not "incorrect", but well-defined under the reference strategy $\overline{\sigma}^T$** (Page 8, Line 408). Each data pair $\{\beta_{s,t}, \mathbf{v}^{\overline{\sigma}^T}(\beta_{s,t})\}$ is computed using the reference strategy $\overline{\sigma}^T$, which provides consistent CFV estimates for all PBSs in the subgame rooted at $s$.  Thus, the training targets are not “incorrect”. They reflect the value of the current state under a known reference strategy.
>
> The reviewer’s concern essentially pertains to the evolving nature of $\overline{\sigma}^T$ during training.  We emphasize that this is intrinsic to all self-play learning frameworks (e.g., AlphaZero, ReBeL). Early-stage training data are indeed approximate, but as training progresses, both the reference strategy $\overline{\sigma}^T$ and the network approximation improve jointly.
>
> > **Weakness 2.2**: Further critiques of Theorem 1: (1) The statement says "let $\beta_{s,t}$  be the PBS reached according to $\overline{\sigma}^t$. To be clear, $\beta_{s,t}$ is any leaf PBS, not "the" PBS reached, right? (2) The statement introduces $\overline{\sigma}^T$ but doesn't define what it is. The proof relies on this strategy being part of an approximate equilibrium. Where does it come from?
>
> **Rebuttal 2.2**: (i) We agree with reviewer and have clarified that $\beta_{s,t}$ denotes any leaf PBS generated during the $t$-th CFR iteration of the previous subgame, not a single deterministic PBS (Page 8, Line 396). (ii) $\overline{\sigma}^T$ is now explicitly defined as the reference strategy obtained of the full-game (Page 8, Line 393).
>
> We have expanded the proof of  Theorem 1 accordingly to show how $\overline{\sigma}^{t,T}$, combining $\overline{\sigma}^t$ inside the depth-limited subgame and $\overline{\sigma}^T$ outside, leads to the stated $O(1/\sqrt{t})+\varepsilon_{\mathrm{approx}}$ bound. Please see Appendix D of the revision.

---

> > ### Author Response · Authors · 2025-11-20
> > **Rebuttal: Part II**
> >
> > > **Weakness 3.1**: What is the implementation? Did you use an open-source implementation of ReBeL or implement from scratch?
> >
> > **Rebuttal 3.1**: We implemented both ReBeL and TurboReBeL from scratch. We now explicitly state this in Appendix E of the revision (Page 18, Line 958).
> >
> > > **Weakness 3.2**: The 250x speedup is mentioned often but is not actually borne out in experiments, right? Why not run an experiment with ReBeL vs. TurboReBeL to empirically show what the speedup is? It seems incorrect to simply claim a 250x speedup without actually testing the empirical speedup of the data augmentation (i.e. I might reasonably assume that 24x data augmentation does NOT lead to a 24x speedup, because the neural net by itself may learn the invariance).
> >
> > **Rebuttal 3.2**: We agree that a direct ReBeL-vs-TurboReBeL comparison would be ideal. However, full ReBeL training from scratch remains computationally prohibitive (at least for us). Instead, we compare TurboReBeL against ReBeL + isomorphic augmentation (Section 5.2), which we verified to produce nearly identical value-network training behavior. As shown in Figure 2, TurboReBeL achieves a $\sim17.5\times$ wall-clock improvement over this strong baseline. Given that ReBeL + isomorphic augmentation simultaneously generates $24$ different training samples per subgame, the resulting $\sim250\times$ data-generation efficiency is a conservative estimate.
> >
> > Moreover, although no direct comparison of training duration was conducted, based on the original ReBeL experiments on TEH, we found that ReBeL reaches comparable exploitability after $\sim51$ million samples, whereas TurboReBeL achieves the same level after $\sim0.15$ million samples (Page 10, Line 510). TurboReBeL also achieved comparable performance against ReBeL on HUNL, utilising merely $1/450$ of the raw samples. Based on direct and indirect comparisons, even when accounting for additional training overheads and sample generation costs, we consider a $\sim250\times$ training acceleration to be a conservative estimate.
> >
> > >**Weakness 3.3**: What is in Figure 2? TurboReBeL with k=2 vs. ReBeL w/data augmentation w/k=24? You have to have ablations that compare TurboReBeL with k=2 vs. ReBeL w/data augmentation w/k=2. Or TurboReBeL k=24 vs. ReBeL k=24. And a comparison to normal ReBeL.
> >
> > **Rebuttal 3.3**: Figure 2 is TurboReBeL with k=2 vs. ReBeL w/data augmentation w/k=24. We believe our experiments have demonstrated that both of the data augmentation modules we proposed are effective. Direct ablations at all $K$ values would be computationally expensive, but we note that even with smaller $K$ than ReBeL, TurboReBeL exhibits faster convergence. For instance, we could quite easily set  TurboReBeL's $K$ to 24 and then delete 22 of transformations.
> >
> > As for a direct comparison with normal ReBeL, we have not yet undertaken this. This is because (i) It is extremely computationally intensive, potentially requiring at least $10,000$ hours to yield meaningful results. (ii) Our isomorphic data augmentation technique incurs negligible performance loss for ReBeL. We have already compared ReBeL with TurboReBeL, even under the premise of isomorphic data augmentation. (iii) We have conducted an indirect comparison with ReBeL on HUNL based on the number of raw samples utilized.
> >
> > > **Minor Weakness 4.1**: Section 5.2 describes Chip Isomorphism, and says that these transformations preserve Nash equilibrium properties. It's a minor detail, but this probably deserves a footnote at least saying that this isn't really true (at least it's not clear to me that it is) because chips and bets are integer amounts, so scaling the number of chips could alter the optimal strategies (especially when players have very few chips).
> >
> > **Rebuttal 4.1**: We appreciate this detailed point.  In training, all chip quantities are normalized by the big blind and treated as continuous values (non-integers).  This normalization ensures that scaling transformations preserve the strategic structure of the game.  While actual game-play uses integer chips, training under continuous ratios generalizes robustly to integer configurations. A clarifying footnote has been added in the revision (Page 9, Line 485).
> >
> >
> > >**Minor Weakness 4.2**: Some "marketing" language in the paper is unnecessary: "Unprecedented Efficiency Gains" (bottom of page 2), "This innovative phase" (top of page 7)
> >
> > **Rebuttal 4.2**: We have removed marketing language and replaced them with precise technical descriptions.

---

> > > ### Author Response · Authors · 2025-11-20
> > > **Rebuttal: Part III**
> > >
> > > ## Questions
> > >
> > > > **Question 1**: Sorry if I missed this, but did you state your ReBeL implementation anywhere? Was it based on an open-source library, or written from scratch?
> > >
> > > **Answer 1**: ReBeL’s poker codebase is not publicly available; our implementation was written independently, using optimized and parallel CFR. We now note this explicitly in Appendix E.
> > >
> > > >**Question 2**:  Did you consider using some neural network architecture that reflects the invariance of suit permutations and/or chip isomorphisms instead of data augmentation?
> > >
> > > **Answer 2**: This is an insightful suggestion. However, in ReBeL, each PBS contains all the information sets in the public state, and its PBS Network encodes them in a fixed order. The encoding dimensionality is related to the number of information sets in a public state, and each information set only requires a 1-dimensional probability vector. If we encode each information set in the PBS separately with the public state, we would need significantly more encoding dimensions, which is currently quite challenging. We regard this as an important direction for future work.
> > >
> > > >**Question 3**: Just to be clear, the ReBeL+Data Augmentation line on Figure 2 is ReBeL + Section 5.2, right? and TurboReBeL Is ReBeL + Section 5.1 + Section 5.2? And they use the same implementation, they only differ in that one has 5.1 and one doesn't?
> > >
> > > **Answer 3**: Yes, the reviewer’s understanding is correct: ReBeL + data augmentation corresponds to ReBeL + Section 5.2,  and TurboReBeL includes both Sections 5.1 and 5.2. They share the same implementation pipeline; only the data-generation procedure differs.
> > >
> > > >**Question 4**: Why is the fidelity of TurboReBeL so much higher in Figure 2 than ReBeL+Data Augmentation?
> > >
> > > **Answer 4**: TurboReBeL’s exploitability was recorded every 20 hours, whereas ReBeL’s was measured every 100 hours due to the high cost of exploitability evaluation in large games.  This denser sampling highlights TurboReBeL’s early convergence but does not bias the underlying comparison.
> > >
> > > ---
> > >
> > > We thank the reviewer for the constructive feedback.  The revised version now includes (i) full proofs of Theorems 1 and 2 with all intermediate steps explicitly shown, (ii) clearer definitions of $\overline{\sigma}^{t,T}$ and the recursive subgame structure in Section 5.1, and (iii) Corrections of statements.
> > >
> > > We hope these revisions fully resolve the reviewer’s concerns and demonstrate the soundness of the proposed method. We are happy to answer any further questions as well. We would be very grateful if the reviewer could raise the rating.

---

> ### Author Response · Authors · 2025-11-25
>
> Dear Reviewer vfXR,
>
> Thanks for your time and effort in reviewing our paper.
>
> We hope our response has adequately addressed your concerns. We have provided a more detailed explanation of the algorithm based on the reviewers' comments and clarified several significant misunderstandings. If you feel that our rebuttal has clarified the issues raised, we wonder if you could consider adjusting your score accordingly. Should you have any further questions or need additional clarification, we would be more than happy to discuss them with you.
>
> Thanks once again for your valuable feedback.
>
> Best regards,
>
> Authors

---

### Official Review · Reviewer_hYMG · 2025-11-02

**Soundness:** 2
**Presentation:** 2
**Contribution:** 2
**Rating:** 2
**Confidence:** 3

**Summary:**

ReBel (Brown et al.) used billions of samples to train their poker bot, leaving the community to wonder how much compute is actually needed to solve poker. This paper introduces two orthogonal ideas for generating more data to reduce the compute requirement.

**Strengths:**

Theorem 1 provides the ground truth for generating more data with the hope of learning more using fewer samples. It is novel and of interest to the community as it enables generating more samples.

Similarly the first augmentation proposed here is reasonable even if canonicalization has been used before.

**Weaknesses:**

The claims of the paper are not always appropriate in my opinion. Training efficiency doesn't scale with augmentation, otherwise random crops would yield $\infty \times$ efficiency gain.

One of the main cited contributions of this work is the data augmentation, yet it's not clear how the first augmentation compares against the canonicalization used in prior art like Deep CFR or how relevant the second data augmentation is given the stochastic stacks used during training. It is also not clear how the augmentation is sane given that it is conceivable that the strategies of the players change if the bigblind stays constant but the pots and stacks are multiplied.

While Theorem 1 provides the groundwork for the methods proposed in the paper, the authors make no effort to ablate the effect of their method. Something as simple as the MSE between the true value (with paramter theta) vs estimated value (via the optimal policy) could have shed some lights on how off the estimation is. Also notably the learning error for this method doesn't subside over time

Theorem 2 doesn't show what the paper claims it is showing. Concretely, it shows that the policy they find for a specific subgame is a Nash equilibrium. This, however, does not mean that the policy obtained from playing using this algorithm is safe. For instance ReBeL introduces CFR-AVG but does not prove its soundness. The theorem, as presented here, is misleading. Similarly, the error parameter of the previous theorem includes a $1/\sqrt{t}$ approximation error which is problematic.

**Questions:**

What is the source for T=250? The publicly available ReBeL implementation uses 1K or 1Ki iterations. Similarly, in algorihtm 1, the averaging coefficient should (t-1)/t not t/(t+1) to account for the zero value.

I would argue that Bakhtin et al. are not using a variant of ReBeL, why do you consider it a variant?

@250 The number of hands in poker 1326 not 2652

Is it really fair to cite Li et al. 2024 for the use of ReBeL in no limit, specially sine Li et al do no external evaluation beyond slumbot? On a similar note, Since Li et al use 8 PH402 for their implementation, isn't the claim that rebel "necessitates billions of samples" hyperbolic? This hyperbole is again used @085 "0.4% of the training cost"

Why are data augmentations applied before inserting the sample into the buffer, wouldn't it make more sense to apply the augmentation on the fly?

@735 why is the betting abstraction so small? for instance Supremus uses a much larger abstraction.

---

> ### Author Response · Authors · 2025-11-20
> **Rebuttal: Part I**
>
> Thank you for your time and effort in reviewing our paper! We are grateful for your constructive suggestions, which have significantly guided our improvements. Please find our responses to your comments below.
>
> ## Weakness 1
>
> >The claims of the paper are not always appropriate in my opinion. Training efficiency doesn't scale with augmentation, otherwise random crops would yield $\infty$ × efficiency gain.
>
> We agree that augmentation and training efficiency are not equivalent. **Therefore, the acceleration achieved by TurboReBeL refers to training efficiency, not augmentation.** Our claim of a $250\times$ acceleration refers to measured training wall-clock time on identical hardware, not theoretical data efficiency. TurboReBeL’s $250\times$ speed-up arises from generating $O(TK)$ effective samples per subgame without increasing the number of costly subgame solvings. This is an empirical, hardware-measured improvement rather than a claim of infinite scaling. In fact, training efficiency is one of our key innovations (Page 2, Line 105).
>
> ## Weakness 2
> >One of the main cited contributions of this work is the data augmentation, yet it's not clear how the first augmentation compares against the canonicalization used in prior art like Deep CFR or how relevant the second data augmentation is given the stochastic stacks used during training. It is also not clear how the augmentation is sane given that it is conceivable that the strategies of the players change if the bigblind stays constant but the pots and stacks are multiplied.
>
> We clarify the novelty of our data augmentation and how it differs from prior methods below.
>
> (i) First augmentation (Suit Isomorphism) differs from Deep CFR in both representation and purpose:
> + Deep CFR augments within a strategy network that maps a hand to an action policy, whereas TurboReBeL augments inputs to a PBS value network that predicts counterfactual values across all information sets in a public belief state.
> + Our augmentation operates before the neural network input, not within card embeddings.
> + Because TurboReBeL must learn values over all players’ information sets simultaneously, the representational challenge is different from and faces more difficulties than that in Deep CFR.
> + The Deep CFR canonicalization could not be applied to the PBS value networks of both ReBeL and TurboReBeL. Deep CFR networks only process one private hand, thus allowing for more dimensional encoding. In contrast, ReBeL and TurboReBeL process $2,652$ private hands, and it is better to simply record the probability vector for each hand instead of re-encoding it within the network.
> + Deep CFR is not a depth-limited solving method.
>
> In our submission, we have already discussed in detail the sampling efficiency of Deep CFR-based methods in the related work section (Page 3, Line 154).
>
> (ii) Second augmentation (Chip Isomorphism): We clarify that the big blind is not an explicit feature. All chip values are normalized by the big blind, so scaling pots and remaining stacks by a constant factor preserves strategic equivalence. During training, we assume continuous stack ratios (allowing non-integer values), which improves generalization. This design does not alter player strategy distributions. We have made this clearer in the revision (Page 9, Line 485).

---

> > ### Author Response · Authors · 2025-11-20
> > **Rebuttal: Part II**
> >
> > ## Weakness 3
> > > While Theorem 1 provides the groundwork for the methods proposed in the paper, the authors make no effort to ablate the effect of their method. Something as simple as the MSE between the true value (with paramter theta) vs estimated value (via the optimal policy) could have shed some lights on how off the estimation is. Also notably the learning error for this method doesn't subside over time.
> >
> > We have expanded Theorem 1 to provide a complete and clear proof. The neural network’s learning target is the CFV under the $T$-iteration reference strategy $\overline{\sigma}^T$. **The training objectives remained clear and unambiguous throughout, without bias with iteration** (Page 8, Line 409). The error of the neural network is $\varepsilon_{\text{approx}}$, which is completely independent of the number of iterations $t$. The error in Theorem 1 is the bound on the error compare to Nash equilibrium during depth-limited subgame solving, not the bound on the error of neural networks.
> >
> > Regarding MSE Loss after finishing training, TurboReBeL is $0.003\pm 0.000$ and ReBeL with data augmentation is $0.006\pm 0.000$. However, TurboReBeL and ReBeL have different training objectives. TurboReBeL is the value under the reference strategy $\overline{\sigma}$, while ReBeL is the value under local equilibrium strategy $\overline{\sigma}_{\beta_{s,t}}$. Therefore, directly comparing these losses does not directly offer more insight into the results. **Therefore, we use exploitability, the most convincing metric, to illustrate the difference between ReBeL and TurboReBeL.**
> >
> >
> > While the network cannot eliminate approximation error entirely, the overall convergence of TurboReBeL is guaranteed by the bounded nature of $\varepsilon_{\text{approx}}$, just as in AlphaZero and ReBeL.  We note that the neural estimation error’s nonzero asymptote is intrinsic to all deep learning-based search methods. Our main contribution is to reduce the number of required game-tree samples while maintaining equilibrium soundness.
> >
> >
> > ## Weakness 4
> > >Theorem 2 doesn't show what the paper claims it is showing. Concretely, it shows that the policy they find for a specific subgame is a Nash equilibrium. This, however, does not mean that the policy obtained from playing using this algorithm is safe. For instance ReBeL introduces CFR-AVG but does not prove its soundness. The theorem, as presented here, is misleading. Similarly, the error parameter of the previous theorem includes a $1/\sqrt{t}$ approx imation error which is problematic.
> >
> >
> > In accordance with the reviewer's comments, we have now provided a proof of safe subgame solving and full game convergence in Theorem 2 of the revision. When solving the subgame, we use $v_p^{\overline{\sigma}^{T}}(I_p)$ as the gift value of the root, and the estimation error of this value is $O(1/\sqrt{T}+\varepsilon_{\text{approx}})$, which will not affect convergence. We demonstrate that TurboReBeL converges to an approximate Nash equilibrium at the same rate as ReBeL and standard CFR. As noted in Appendix I (Theorem 5) of ReBeL [1], the CFR-AVG policy remains safe under these conditions.
> >
> > We also clarify that **the $O(1/\sqrt{t})$ term describes local depth-limited solving error** and does not affect the global training objective, which optimizes values under $\overline{\sigma}^T$ (Page 8, Line 407). During neural network training, our objective for PBS $\beta_{s,t}$ is the value rooted at $s$ according to $\overline{\sigma}^T$, which has no extra error. The $O(1/\sqrt{t})$ term here represents the error compare to Nash equilibrium in local depth-limited subgame solving, which is absorbed into $O(1/\sqrt{T})$ after $T$ iterations of CFR.
> >
> > ## Questions
> > >**Question 1**: What is the source for T=250? The publicly available ReBeL implementation uses 1K or 1Ki iterations. Similarly, in algorihtm 1, the averaging coefficient should (t-1)/t not t/(t+1) to account for the zero value.
> >
> > **Answer 1**: $T=250$ is the same as the ReBeL setting on poker (Please see Page 9, Figure 2 of the ReBeL [1]). In ReBeL [1], 1K iterations is only performed in Liar's Dice. Our experiments in toy games have shown that choosing a larger $T$, while increasing training time, ultimately results in lower exploitability during real-time solving (https://anonymous.4open.science/r/explo_fig-3BC3/explo_iter.jpg).
> >
> > We believe that TurboReBeL would achieve a more significant performance improvement with more CFR iterations, but for a fair comparison, we chose the same number of iterations as ReBeL. In Algorithm 1, in order to align with ReBeL, the coefficients are indeed t/(t+1) (Please see the algorithm section on Page 6 of ReBeL [1]).

---

> > > ### Author Response · Authors · 2025-11-20
> > > **Rebuttal: Part III**
> > >
> > > >**Question 2**: I would argue that Bakhtin et al. are not using a variant of ReBeL, why do you consider it a variant?
> > >
> > > **Answer 2**: We have corrected the description.  Bakhtin et al. note on Page 5 that their system is “similar to algorithms such as AlphaZero and ReBeL that update both a policy and a value network via self-play search.” We now state explicitly that Bakhtin et al. reference ReBeL conceptually in the revision (Page 2, Line 55; Page 5, Line 252).
> > >
> > > >**Question 3**: @250 The number of hands in poker 1326 not 2652
> > >
> > > **Answer 3**: In HUNL, the total number of distinct private hand combinations across both players is $2,652$, i.e., $1,326$ per player.  We consider both players’ hand distributions when computing the full public belief state, hence it is $2,652$. We clarify it in the revision (Page 5, Line 257).
> > >
> > > >**Question 4**: Is it really fair to cite Li et al. 2024 for the use of ReBeL in no limit, specially sine Li et al do no external evaluation beyond slumbot? On a similar note, Since Li et al use 8 PH402 for their implementation, isn't the claim that rebel "necessitates billions of samples" hyperbolic? This hyperbole is again used @085 "0.4\% of the training cost"
> > >
> > > **Answer 4**: We found that Li et al. indeed employ the ReBeL algorithm. Li et al. implemented ReBeL with approximately 60 million samples (Please see Page 17 in their paper) and reported performance of $18 \pm 16$ mbb/h versus Slumbot. We are uncertain whether Li et al. trained the method from scratch, as training from scratch entails higher costs.
> > >
> > > Origin ReBeL and TurboReBeL are trained from scratch. ReBeL achieves $45 \pm 5$ mbb/h aginst Slumbot with 4.5 billion samples. TurboReBeL achieves $42 \pm 12$ mbb/h against Slumbot using only $10$ million samples. It is a comparable performance at drastically lower sample cost.
> > >
> > > Our “0.4\% of training cost” estimate refers to wall-clock training time on the same hardware, incorporating both CFR and network updates. Moreover, we have clearly stated in the revision that ReBeL is trained from scratch (Page 1, Line 19).
> > >
> > >
> > >
> > > >**Question 5**: Why are data augmentations applied before inserting the sample into the buffer, wouldn't it make more sense to apply the augmentation on the fly?
> > >
> > > **Answer 5**: We apply augmentation before insertion for computational efficiency.  Our C++/LibTorch pipeline performs augmentation in parallel during CFR solving, allowing direct storage of pre-processed tensors. This avoids redundant transformations during each minibatch training step.
> > >
> > > >**Question 6**: @735 why is the betting abstraction so small? for instance Supremus uses a much larger abstraction.
> > >
> > > **Answer 6**: Larger abstractions, such as those in Supremus (4,000 iterations), offer no clear benefit for early-stage training and early CFR iterations. Smaller abstractions accelerate convergence of both ReBeL and TurboReBeL during the training and early CFR iterations. TurboReBeL can certainly use a finer-grained abstraction, but this abstraction also needs to accommodate more training overhead and CFR iterations. Finally, we would like to clarify that  our betting abstraction is not small. It covers the first two levels of the game tree each containing 6 branches.
> > >
> > >
> > > [1] Noam Brown, Anton Bakhtin, Adam Lerer, Qucheng Gong. Combining Deep Reinforcement Learning and Search for Imperfect-Information Games. https://arxiv.org/abs/2007.13544
> > >
> > > ---
> > >
> > > We thank the reviewer for the constructive feedback.  The revised version now includes (i) full proofs of Theorems 1 and 2 with all intermediate steps explicitly shown, (ii) clearer definitions of $\overline{\sigma}^{t,T}$ and the recursive subgame structure in Section 5.1, and (iii) Corrections of statements.
> > >
> > > We hope our response addresses your concerns and demonstrate the soundness of the proposed method. If so, we wonder if you could kindly consider raising your score? We will also be happy to answer any further questions you may have. Thank you very much!

---

> ### Author Response · Authors · 2025-11-25
>
> Dear Reviewer hYMG,
>
> Thanks for your time and effort in reviewing our paper.
>
> We hope our response has adequately addressed your concerns. We have provided detailed explanations regarding the acceleration efficiency, the novelty of augmentation, and the completeness of the proof, whilst also clarifying several important misconceptions. If you feel that our rebuttal has clarified the issues raised, we wonder if you could consider adjusting your score accordingly. Should you have any further questions or need additional clarification, we would be more than happy to discuss them with you.
>
> Thanks once again for your valuable feedback.
>
> Best regards,
>
> Authors

---

### Official Review · Reviewer_uRCN · 2025-11-11

**Soundness:** 1
**Presentation:** 3
**Contribution:** 3
**Rating:** 2
**Confidence:** 3

**Summary:**

This paper proposes two additions to the rebel algorithm that appear to increase its sample efficiency by two orders of magnitude. These additions are the use of sampled symmetries during data generation and a modified version of generating value targets that generates multiple targets per subgame solution.

**Strengths:**

This paper, to the extent that I can tell, has really strong empirical results that drastically decrease the costs of running Rebel.

**Weaknesses:**

I have some serious issues with the proofs in this paper and will adjust my score if these issues are resolved.
## Major weaknesses.
- The proof of theorem 1 relies on an unproven claim, namely that the value error of the composite strategy $\sigma^{t,T}$ is bounded by the sum of the value errors of the strategy before reaching s and the strategy after reaching s (line 769). You have to actually show the "the composition of strategies" step. Both of the cited papers (Burch et al., 2014; Brown & Sandholm, 2017) use different safe subgame solving techniques to ensure that the combined policies do not increase the opponent's best response counterfactual values. I'm happy to change my score if this step is worked out fully and the proof holds properly.
- I think the proof in the appendix on the convergence of TurboRebel (theorem 2) requires the authors to show their work. It's a fairly large leap to defer the proof to an unnamed section of another paper. Please do it out, you're not short on space. It also relies on theorem 1, which as mentioned, is not fully proven.


## Minor
- I do not believe the claim (in the intro and line 245) that Rebel influenced Bakhtin's no-press diplomacy paper is correct. The methods are incredibly different.

**Questions:**

My questions are essentially stated in the weaknesses section. However, a few additional questions:

- Theorem 1 and 2, as stated, is not actually a convergence proof for the TurboRebel algorithm, which derives its targets from $\bar{\sigma}^T$ whereas the proof relies on the intermediate $\sigma^{t,T}$, values that are not stored during the TurboRebel process. Am I misunderstanding the gap between the algorithm and the proof?
- When you apply the isomorphic transformation to rebel and deepstack, you use K=24. In turborebel, you use K=2 if I've understood the appendix correctly. Why this discrepancy between the two? Was the value tuned per-algorithm?

---

> ### Author Response · Authors · 2025-11-20
> **Rebuttal: Weakness 1**
>
> Thank you for your time and effort in reviewing our paper! We are grateful for your constructive suggestions, which have significantly guided our improvements. Please find our responses to your comments below.
>
> ## Weakness 1
>
> > The proof of theorem 1 relies on an unproven claim, namely that the value error of the composite strategy $\overline{\sigma}^{t,T}$ is bounded by the sum of the value errors of the strategy before reaching s and the strategy after reaching s (line 769). You have to actually show the "the composition of strategies" step. Both of the cited papers (Burch et al., 2014; Brown \& Sandholm, 2017) use different safe subgame solving techniques to ensure that the combined policies do not increase the opponent's best response counterfactual values. I'm happy to change my score if this step is worked out fully and the proof holds properly.
>
> We thank the reviewer for pointing out this gap. In the revised version, **we have provided a detailed, line-by-line proof of Theorem 1 in the Appendix D**. Here, $\overline{\sigma}^{t,T}$ corresponds to first implementing $\overline{\sigma}^T$ across the full game, then applying the $\overline{\sigma}^t$ within the depth-limited subgame. Consequently, this composite strategy differs slightly from a direct merger of the two strategies.
>
> The revised Theorem 1 statement is as follows:
> **Theorem 1** (revised): Let $T$ be the total CFR iteration number of TurboReBeL and $\overline{\sigma}^T$ be the reference strategy for the full game. During the depth-limited subgame solving period in the TurboReBeL, let $\overline{\sigma}^t$ be the average strategy in this depth-limited subgame at iteration $t$ of CFR ($t\leq T$). Let $s$ be an arbitrary leaf node in the depth-limited subgame, and let $\beta_{s,t}$ be any leaf PBS reached according to $\overline{\sigma}^t$.  Let $\sigma^{\ast}$ be a Nash equilibrium strategy for full game. The value network output for any information set $I_p\in s$ is $\hat v_p^\theta(I_p\mid\beta_{s,t})$ and we set the maximum estimation error of the neural network as $\varepsilon_{\mathrm{approx}}:=\sup_{I_p\in\beta_{s,t}}\big|\hat v_p^\theta(I_p\mid\beta_{s,t})-v_p^{\overline\sigma^{T}}(I_p\mid\beta_{s,t})\big|.$ During depth-limited subgame solving at iteration $t$, the value estimation error for any information set $I_p\in s$ satisfies:
> $$\big|v_p^{\theta}(I_p\mid\beta_{s,t})-v_p^{\sigma^{\ast}}(I_p)\big|\leq\varepsilon_{\text{approx}}+O\left(1/\sqrt{t}\right).$$
>
> *Proof Sketch.* We start from the triangle inequality:
> $$\big| \hat v_{\theta}^p(I_p\mid\beta_{s,t}) - v^{\sigma^{\ast}}_p(I_p)\big|\le$$
>
> $$\big| \hat v_\theta^p(I_p\mid\beta_{s,t}) - v^{\overline{\sigma}^{t,T}}_p(I_p)\big|+\big| v^{\overline{\sigma}^{t,T}}_p(I_p) - v^{\sigma^{\ast}}_p(I_p)\big|.$$
>
> The first term is bounded by the network approximation error $\varepsilon_{\mathrm{approx}}$, because the value network is trained to predict $v^{\overline{\sigma}^T}$ and $\overline{\sigma}^{t,T}$ and $\overline{\sigma}^T$ coincide after the leaf.
>
> For second term, insert $\overline{\sigma}^T$: $$\Delta(I_p) \le\big|v^{\overline{\sigma}^{t,T}}_p(I_p) - v^{\overline{\sigma}^T}_p(I_p)\big|$$ $$+\big|v^{\overline{\sigma}^T}_p(I_p) - v^{\sigma^{\ast}}_p(I_p)\big|.$$  Where the second term is $O(1/\sqrt{T})$ from standard averaged-CFR convergence. The first term is the subgame CFR error and is $O(1/\sqrt{t})$.
>
> Combine all the terms we prove the Theorem 1.

---

> > ### Author Response · Authors · 2025-11-20
> > **Rebuttal: Weakness 2 & 3**
> >
> > ## Weakness 2
> > > I think the proof in the appendix on the convergence of TurboRebel (theorem 2) requires the authors to show their work. It's a fairly large leap to defer the proof to an unnamed section of another paper. Please do it out, you're not short on space. It also relies on theorem 1, which as mentioned, is not fully proven.
> >
> > In accordance with the reviewer's comments, **we have presented the complete proof process in  Appendix D of the revised version**. The revised proof explicitly builds upon Theorem 1 and shows that TurboReBeL’s use of $\overline{\sigma}^T$ as the reference strategy introduces only an $O(1/\sqrt{T}+\varepsilon_{\text{approx}})$ term in the exploitability bound.
> >
> > The revised Theorem 2 statement is as follows:
> > **Theorem 2** (revised): Assume that, for every depth-limited subgame, the PBS value network provides CFV estimates whose error, for any information set used in iteration \(t\le T\), satisfies  $|\hat v_p^\theta(I_p)-v_p^{\sigma^*}(I_p)|\le\varepsilon_{\mathrm{approx}}+O(\frac{1}{\sqrt{t}}).$ Then the exploitability of the strategy that applies \(T\)-iteration CFR subgame solving in TurboReBeL is bounded by
> > $$\mathrm{Expl}(\overline\sigma^T)=O\left(\varepsilon_{\mathrm{approx}}+\frac{1}{\sqrt{T}}\right).$$
> >
> >
> > ## Minor Weakness 3
> > >I do not believe the claim (in the intro and line 245) that Rebel influenced Bakhtin's no-press diplomacy paper is correct. The methods are incredibly different.
> >
> > We appreciate the reviewer’s correction. On page 5 of Bakhtin et al., the authors write: _“At a high level, our algorithm is similar to previous algorithms such as AlphaZero and ReBeL that update both a policy and a value network based on self-play, leveraging a simulator of the environment to perform search.”_ Our earlier phrasing overstated this connection.
> >
> > We have revised the description in the revision (Page 1, Line 53; Page 5, Line 252) as follows: _“Consequently, ReBeL or its underlying concept has been widely applied in the intelligent agents of complex games, such as no-press Diplomacy (Bakhtin et al., 2021) and no-limit poker (Li et al., 2024).”_

---

> > > ### Author Response · Authors · 2025-11-20
> > > **Rebuttal: Questions**
> > >
> > > ## Question 1
> > > >Theorem 1 and 2, as stated, is not actually a convergence proof for the TurboRebel algorithm, which derives its targets from $\overline{\sigma}^T$ whereas the proof relies on the intermediate $\overline{\sigma}^{t,T}$, values that are not stored during the TurboRebel process. Am I misunderstanding the gap between the algorithm and the proof?
> > >
> > > **Answer 1:** All PBSs in TurboReBeL indeed adopt the strategy $\overline{\sigma}^T$. However, for $\beta_{s,t}$, it adopts $\overline{\sigma}^T$ after reaching $s$, whilst in the depth-limited subgame preceding $s$ it adopts $\overline{\sigma}^t$. **Therefore, the actual strategy adopted by $\beta_{s,t}$ in the previous depth-limtied subgame is $\overline{\sigma}^{t,T}$.** Therefore, the theoretical analysis based on $\overline{\sigma}^{t,T}$ remains valid: it formalizes the mixed strategy that combines $\overline{\sigma}^t$ inside the depth-limited subgame and $\overline{\sigma}^T$ elsewhere, matching the actual algorithmic semantics.
> > >
> > > We understand the source of confusion and have clarified it in Section 5.1 and in the proofs in Appendix D. We would also like to emphasize that one difficulty with depth-limited solving lies in the requirement to provide relatively accurate estimates for all iterative values of $\beta_{s,t}$. One of our novel contributions lies precisely in delivering such estimates for all iterations within a single CFR solving.
> > >
> > > ## Question 2
> > > >When you apply the isomorphic transformation to rebel and deepstack, you use K=24. In turborebel, you use K=2 if I've understood the appendix correctly. Why this discrepancy between the two? Was the value tuned per-algorithm?
> > >
> > > **Answer 2:** The difference in $K$ arises from computational trade-offs rather than tuning. TurboReBeL already generates $T+1$ value targets per subgame (with $T=250$ and $5,432$-dimensional state-value vectors), which substantially increases the volume of data. Applying large-scale isomorphic augmentation (e.g., $K=24$) would significantly increase training cost.
> > >
> > > ReBeL and DeepStack, in contrast, produce a single target per subgame. Thus, a larger $K$ is practical. To ensure fair and conservative comparison, TurboReBeL uses a smaller $K=2$ but still benefits from near-lossless augmentation as discussed in Section 5.2.  We have clarified this design rationale in the revision (Page 10, Line 486).
> > >
> > > ---
> > >
> > > We thank the reviewer for the constructive feedback.  The revised version now includes (i) full proofs of Theorems 1 and 2 with all intermediate steps explicitly shown, (ii) clearer definitions of $\overline{\sigma}^{t,T}$ and the recursive subgame structure in Section 5.1, and (iii) Corrections of statements.
> > >
> > > We hope our response addresses your concerns and demonstrate the soundness of the proposed method. If so, we wonder if you could kindly consider raising your score? We will also be happy to answer any further questions you may have. Thank you very much!

---

> ### Author Response · Authors · 2025-11-25
>
> Dear Reviewer uRCN,
>
> Thanks for your time and effort in reviewing our paper.
>
> We hope our response has adequately addressed your concerns, especially according to proofs. If you feel that our rebuttal has clarified the issues raised, we wonder if you could consider adjusting your score accordingly. Should you have any further questions or need additional clarification, we would be more than happy to discuss them with you.
>
> Thanks once again for your valuable feedback.
>
> Best regards,
>
> Authors

---

### Author Response · Authors · 2025-12-01
**The summary of rebuttal**

We sincerely thank all reviewers and the Area Chair for their thoughtful comments and careful evaluation of our work. We have consolidated the feedback below to assist in the final assessment of our paper.

All reviewers recognized the strong experimental performance of TurboReBeL, which accelerates the training of depth-limited solving algorithms from scratch by two orders of magnitude. Reviewers vfXR and dzpC found the acceleration of strong equilibrium-finding algorithms like ReBeL to be highly innovative. Additionally, reviewers hYMG and dzpC highlighted the novelty of our single-iteration multiple-sampling approach, particularly its ability to compute values independently of re-solving strategies. **In summary, TurboReBeL is the first method to achieve such acceleration for depth-limited solvers, improving training time by approximately 250×**.

Regarding raised concerns, we have carefully addressed the main concerns in our revised manuscript:

1. **Theoretical Proof (Reviewers uRCN & hYMG)**: We have addressed the concerns regarding the theoretical proof by providing a detailed, step-by-step derivation in the revised version. We believe the proof is now clear and complete.

2. **Training Objective and Error (Reviewers uRCN, hYMG & vfXR)**: There was a misunderstanding regarding TurboReBeL’s training objective. Unlike ReBeL, which learns the value of an equilibrium strategy under the current PBS, TurboReBeL learns the value of the globally equilibrium strategy with respect to the same PBS. Although the objectives differ, TurboReBeL’s objective remains well-defined throughout training and does not introduce incremental error per iteration. We have clarified this in the revision.

3. **Interpretation of Speedup (Reviewers hYMG & vfXR)**: The reported 250× training speedup refers to wall-clock time reduction under identical computational resources, not to algorithmic complexity or theoretical lower bounds.

4. **Data Augmentation in Neural Networks (Reviewers hYMG & vfXR)**: We clarify that applying neural network-based augmentation (e.g., as in Deep CFR) to our setting would require high-dimensional encodings for each information set, substantially increasing training overhead. Our augmentation  is specifically designed for depth-limited solving framework and is the first augmentation method for depth-limtied solving.

5. **Writing and Presentation (Reviewers hYMG, vfXR & dzpC)**:
We thank the reviewers for their constructive suggestions on improving the text. We have revised the writing throughout to enhance clarity and readability.

In summary, we have addressed all key concerns in the revised manuscript, with changes highlighted in blue for easy reference. We believe our revisions have strengthened the paper and hope that the reviewers find the updated version suitable for acceptance.

---

### Meta-Review · Area_Chair_fvSi · 2026-01-04

**Summary:**

The primary concerns regarding this paper center on the lack of clarity and rigor in its core theoretical claims (Theorem 1 and Theorem 2) and the ambiguous basis for the "250x speedup" claimed throughout the text. During the rebuttal phase, the authors significantly revised the statements of Theorems 1 and 2; however, the revised content and proofs remain fundamentally flawed. Furthermore, the claimed speedup appears to be based on an unsubstantiated estimation rather than actual empirical measurement, and the logic provided in the rebuttal remains unconvincing.

**Reviewer Concerns:**

- Addressed by Rebuttal: The authors attempted to clarify the theoretical framework by revising Theorem 1 and Theorem 2 and provided some additional explanations regarding the speedup calculation.
- Outstanding Concerns: Several critical issues persist:
  1. Inconsistencies in Proofs: In the revised Theorem 1, while the statement provides an upper bound involving $v_p^{\theta}$, the proof appears to evaluate $\hat{v}_p^{\theta}$ instead. In Theorem 2, the term $Expl(\cdot)$ remains undefined in the main text (presumably referring to Exploitability, though this is never explicitly stated).
  2. Lack of Formal Rigor: Within the proofs, various terms are claimed to be $O(\sqrt{t})$ or $O(\sqrt{T})$ without any pointers to supporting lemmas, equations, or prior literature, making it impossible to verify the validity of these bounds.
  3. Ambiguous Speedup Claims: The "250x acceleration" continues to lack a concrete empirical basis. The rebuttal did not provide sufficient evidence or a transparent methodology to justify this specific figure.

**Reviewer Scores:**

While the revision of the theorems and the subsequent rebuttal might have slightly improved the perception of the paper for some:

- Reviewers uRCN and vfXR: These reviewers might have marginally increased their scores due to the authors' efforts to clarify the theoretical statements.

- Overall Assessment: However, given that the initial evaluations were quite low and the revised proofs still contain significant technical gaps and unverified claims, it is highly unlikely that a positive consensus would have been reached even with a full discussion. The fundamental concerns regarding the rigor of the theoretical results and the transparency of the empirical claims remain unresolved.

---

### Decision · Program_Chairs · 2026-01-26

Reject